# Improving Soft Unification with Knowledge Graph Embedding Methods

Xuanming Cui [1]    Chionh Wei Peng [2]    Adriel Kuek [2]    Ser-Nam Lim [1]

## Abstract

Neural Theorem Provers (NTPs) present a promising framework for neuro-symbolic reasoning, combining end-to-end differentiability with the interpretability of symbolic logic programming. However, optimizing NTPs remains a significant challenge due to their complex objective landscape and gradient sparcity. On the other hand, Knowledge Graph Embedding (KGE) methods offer smooth optimization with well-defined learning objectives but often lack interpretability. In this work, we propose several strategies to integrate the strengths of NTPs and KGEs, and demonstrate substantial improvements in both accuracy and computational efficiency. Specifically, we show that by leveraging the strength of structural learning in KGEs, we can greatly improve NTPs' poorly structured embedding space, while by substituting NTPs with efficient KGE operations, we can significantly reduce evaluation time by over $1000\times$ on large-scale dataset such as WN18RR with a mild accuracy trade-off.

## 1. Introduction

Deep Learning (DL) methods have recently achieved tremendous progress in various tasks such as language modeling (Touvron et al., 2023; Liu et al., 2023) and content generation (Rombach et al., 2022; Kerbl et al., 2023). However, when compared with symbolic systems, they are still limited by the long-lasting problems of the lack of interpretation, out-of-domain generalizability and reasoning abilities.

To address the above challenges, the concept of Neuro-Symbolic AI (NeSy) has been proposed to integrate DL and symbolic AI into one end-to-end differentiable system. A popular approach for such integration is to embed discrete symbols into continuous vector space to enable end-to-end

[1]Department of Computer Science, University of Central Florida [2]DSO National Laboratories. Correspondence to: Xuanming Cui <xuanming.cui@ucf.edu>.

*Proceedings of the 42$^{nd}$ International Conference on Machine Learning*, Vancouver, Canada. PMLR 267, 2025. Copyright 2025 by the author(s).

differentiability (Rocktäschel & Riedel, 2017; Minervini et al., 2019; Badreddine et al., 2022). Neural Theorem Prover (Rocktäschel & Riedel, 2017) (NTP) is a representative of such approach. It introduces the concept of soft unification during backward chaining process, where the unification operation is on the learnt embedding space instead of between discrete symbols. Subsequent works Greedy Neural Theorem Prover (GNTP) (Minervini et al., 2019) and Conditional Theorem Prover (CTP) (Minervini et al., 2020) implement top-$k$ rule retrieval and rule reformulation to improve NTP's scalability and performance.

Although there are numerous NeSy frameworks (Cohen, 2016; Das et al., 2018; Minervini et al., 2017) and knowledge-base reasoning methods, we are particularly interested in NTPs due to their straightforward reasoning process based on similarity. This simplicity brings better scaling potential compared to complex tensor operations employed in other NeSy frameworks and GNN-based approaches. Moreover, such similarity-based design also makes NTPs well-suited for leveraging pre-trained embeddings obtained from the recent advances in foundational models, thereby enabling multi-modal reasoning and potential zero-shot capabilities. This is crucial, as most existing NeSy (Maene & De Raedt, 2023; Yang et al., 2017; Minervini et al., 2017) and Knowledge-Graph-related methods (Trouillon et al., 2016; Han et al., 2023; Zhu et al., 2021) struggles to generalize to unseen entities, significantly lagging behind the recent foundational and language models.

Although NTP has been shown to be effective on various datasets, it is known to be hard to optimize (Rocktäschel & Riedel, 2017; Minervini et al., 2019; Maene & Raedt, 2023; de Jong & Sha, 2019). Specifically, as NTPs perform top-k retrievals for unification, only a fraction of retrieved embedding parameters will receive gradient updates. The model optimization is thus heavily dependent on the initialization, and can get stuck in local minima (de Jong & Sha, 2019). DeepSoftLog (Maene & Raedt, 2023) addresses the above limitation by using differentiable probabilistic semiring instead of fuzzy semiring, along with other proposed properties to smooth out the back-propagation process. However, as it requires additional modules for knowledge compilation (Darwiche, 2011) and requires all possible proofs to be considered during training (as opposed to $k$-best approximation), it is intrinsically hard to scale to larger datasets.

Instead of trying to improve NTPs from the sparse gradient perspective which may inevitably face performance-efficiency trade-off, in this paper we take a different angle and tackle the problem from the embedding perspective. We find the NTPs' embedding space is extremely poorly-structured, due to the sparse unification during training (detailed discussion in Section 7). On the other hand, Knowledge Graph Embedding (KGE)s (Bordes et al., 2013; Trouillon et al., 2016) are strong structure learners with well-defined objectives and smooth optimization process. However, as KGEs are purely sub-symbolic algorithms, they lack the interpretability as compared to NTPs.

Motivated by the complementary properties of NTPs and KGEs, we conduct the first systematic study for integrating KGEs into the NTP framework. The rest of the paper is arranged as follow: in Section 3 we provide brief introduction for NTPs and KGEs, and discuss the deficit of NTPs from embedding perspective 7; In Section 8 we explain four strategies for the integration, and conduct detailed experiments in Section 9. Finally, we provide ablation studies to examine important components during the integration. We wish our work can serve as a first step to future studies on such integration and to improve upon existing NTPs.

**Contribution:**

1. We provide the first systematic study for the integration of KGEs into NTPs and propose four integration strategies, improving both performance and efficiency. Note that using KGE as auxiliary was originally proposed in (Rocktäschel & Riedel, 2017). However, it was only briefly mentioned without any further exploration, and is not used in any subsequent NTPs (Minervini et al., 2019; 2020; Maene & De Raedt, 2023).

2. We show that the integration noticeably improve the baseline NTP by a large margin and achieve SOTA results on multiple datasets. In particular, we analyze the issue of NTPs from their learned embedding space perspective and show that it can be benefited from the integration of KGEs. We also show that by leveraging the properties of KGEs we could drastically improve the inference and evaluation efficiency of NTPs.

3. We provide detailed ablations to examine the key factors in the integration. Interestingly, we find NTPs can achieve superior results with pure KGE objectives under several datasets, suggesting the synergy between the two distinct methods.

## 2. Related Work

**Differentiable Logic Programming** algorithms can be roughly divided into two categories. (1) Disentangled perception+reasoning (Manhaeve et al., 2018; Huang et al.,

2021; Yang et al., 2023). This line of works train a neural network to output a probability distribution over symbols, which is then consumed by a differentiable logic solver. For example, DeepProbLog (Manhaeve et al., 2018) guides a neural network with probabilistic circuits constructed by Sentential Decision Diagram (Darwiche, 2011) (SDD). Scallop (Huang et al., 2021) scales up DeepProbLog by only considering top-k possible worlds. NeurASP (Yang et al., 2023) adopts the same strategy, but replace SDD with a Answer Set Programming solver. Under this regime, the neural component is completely separated from the reasoning module. (2)Soft logic programming (Cohen, 2016; Badreddine et al., 2022; Yang et al., 2017; Rocktäschel & Riedel, 2017). This line of works are a continuous relaxation on top of logic programming, by learning a mapping from symbols and logic operations into latent embeddings and differentiable tensor operations. Logic Tensor Network (Badreddine et al., 2022) extends First-Order Logic (FOL) with fuzzy semantics. NEURALLP (Yang et al., 2017) is a rule-based algorithm that extends TensorLog (Cohen, 2016) by learning to soft select and compose rules. Neural Theorem Prover (Rocktäschel & Riedel, 2017) extends backward chaining algorithm with soft unification. Greedy Neural Theorem Prover (Minervini et al., 2019) and Conditional Theorem Prover (Minervini et al., 2020) improve the scalability of NTP by top-k retrieval and soft rule reformulation.

**Knowledge Graph Embedding** (KGE) are SOTA methods for link prediction tasks over large-scale KGs. TRANSE (Bordes et al., 2013) and its extensions (Wang et al., 2014; Xiao et al., 2015) are translation-based KGEs which minimize distance between subject and object, translated by the predicate. On the other hand, RESCAL (Nickel et al., 2011), COMPLEX (Trouillon et al., 2016), TUCKER (Balazevic et al., 2019) etc. use multi-linear maps to combine subject, relation and object for score calculation. Besides traditional KGEs, recent advances such as GeKCs (Loconte et al., 2023) reformulate KGEs with valid probabilistic interpretation, LERP (Han et al., 2023) aggregates sub-graph information as entity representation.

**Path-based KG Algorithms** explicitly learn the multi-hop paths over KGs. They can be applied directly on top of KGEs by handling multi-hop relation paths as compositions over embedding space such as in (Lin et al., 2015), or can be formulated as path-searching algorithms, optimized by Reinforcement Learning objectives such as in (Das et al., 2018; Zhu et al., 2023; Lin et al., 2018). Besides, GNN-based approaches (Schlichtkrull et al., 2017; Zhu et al., 2021) have also shown strong performance on link prediction tasks. For example, RGCN (Schlichtkrull et al., 2017) applies GNN to knowledge-graph learning task. NBFNet (Zhu et al., 2021) generalizes the Bellman-Ford algorithm with neural operators under the GNN framework to solve path formulation.

*Figure 1.* Illustration of CTP algorithm with a transitive rule template and depth $= 1$. Given a goal $(s, R, o)$, it first transforms the goal predicate to a list of predicates forming the proof path. Then it takes the known subject $s$ and predicate $R_i$ to predict the latent object $z_i$ with top-$k$ retrieval; it then uses the predicted $z_i$ as the next subject and predict $z_{i+1}$ to step through the proof path.

## 3. Background

## 4. Neural Theorem Prover

In this section we define the syntax and briefly introduce the SLD resolution and NTP algorithm. We refer the reader to (Rocktäschel & Riedel, 2017) for an in-depth explanation.

**Syntax.** A term $t$ can be either a constant $c$ or a variable $\mathbf{X}$[1]. An atom is defined as a combination of a predicate symbol and a list of terms. Rules are in the form of $\mathbf{H} :\!- \mathbb{B}$, where the head $\mathbf{H}$ is an atom, and the body $\mathbb{B}$ is a list of atoms connected by conjunctions. A rule with no free variables is called a ground rule, and a ground rule with an empty body is called a fact. A substitution, denoted as $\psi = \{\mathbf{X}_1/t_1, \ldots, \mathbf{X}_N/t_N\}$, assigns variables $\mathbf{X}_i$ to terms $t_i$, and applying a substitution to an atom replaces each occurrence of $\mathbf{X}_i$ with the corresponding term $t_i$. In this work, we only consider atoms with binary predicates in the form of $(s, r, o)$, where $s$, $r$ and $o$ denote subject, predicate (relation) and object respectively.

**Backward Chaining.** Given the goal, backward chaining works backward to find supporting facts and rules from the Knowledge Base (KB). It can be seen as an iterative process of applying AND/OR: the OR operation looks for all rules with matching head to perform unification. The AND module is subsequently called to iteratively prove all atoms in the unified rule's body, where the OR module is again called recursively.

**NTP and Soft Unification.** NTPs provide a continuous relaxation of backward chaining by introducing *soft unification*. It calculates a unification score $\gamma = \phi_{\text{NTP}}(c_i, c_j)$ over the embeddings of two symbols, where $\phi_{\text{NTP}}$ refers to the predefined similarity function, $c_i$ and $c_j$ denotes two constant terms to be unified. In case of NTP, a Gaussian kernel is adopted for $\phi_{\text{NTP}}$. The unification score $\gamma$ at each proof state are then aggregated following the min/max fuzzy semiring, also known as the Gödel $t$-norm. Specifically, the AND module performs min aggregation as all sub-goals have to be proved for the given rule, and OR perform max aggregation, since we only need one proof to be true to prove the goal. During training, given a KG $\mathcal{G}$, each fact $(s, r, o) \in \mathcal{G}$ is corrupted to obtain negative samples $(s', r, o)$, $(s, r, o')$

---

[1]We focus on function-free First Order Logic, and therefore does not consider structured terms.

---

and $(s', r, o') \notin \mathcal{G}$. The objective is then defined as the negative log likelihood of the aggregated unification score:

$$\mathcal{L}_{\text{NTP}\,\theta}^{\mathcal{G}} = \sum_{((s,r,o),y)\,\in\,\mathcal{G}} -y \log(\text{NTP}_\theta^{\mathcal{G}}((s,r,o)) -$$
$$(1-y)\log(1 - \text{NTP}_\theta^{\mathcal{G}}((s,r,o)) \quad (1)$$

where $\text{NTP}_\theta^{\mathcal{G}}$ denotes NTP with KG $\mathcal{G}$, parameterized by $\theta$.

## 5. KGs and Embedding Methods

A Knowledge Graph (KG) $\mathcal{G}$ is a directed multi-graph, represented as a collection of triplets (facts) $(s, r, o) \subseteq \mathcal{E} \times \mathcal{R} \times \mathcal{E}$, where $\mathcal{E}$ and $\mathcal{R}$ denote the set of entities and relations in $\mathcal{G}$. A KGE model defines a function that maps triplets to scores $\phi_{\text{KGE}} : \mathcal{E} \times \mathcal{R} \times \mathcal{E} \to \mathbb{R}$. This score function $\phi_{\text{KGE}}$ can be translation-based as in TRANSE (Bordes et al., 2013): $\phi_{\text{TRANSE}}(\boldsymbol{s}, \boldsymbol{r}, \boldsymbol{o}) = -||\boldsymbol{s} + \boldsymbol{r} - \boldsymbol{o}||$, or similarity-based using a multi-linear function (Trouillon et al., 2016; Yang et al., 2015). For instance, COMPLEX (Trouillon et al., 2016) defines the score function as $\phi_{\text{COMPLEX}} = \text{Re}(\langle \boldsymbol{s}, \boldsymbol{r}, \overline{\boldsymbol{o}} \rangle)$, where $\langle \cdot, \cdot, \cdot \rangle$ denotes the tri-linear product, $\text{Re}$ denotes the real part of the complex number, and $\overline{\cdot}$ denotes the complex conjugate. KGEs are traditionally interpreted as energy-based models (EBMs), where the score is interpreted as the negative energy of triplets, and are trained with contrastive objectives and negative log likelihood loss, similar to $\mathcal{L}_{\text{NTP}}$. Besides treating KGEs as EBMs, existing works (Joulin et al., 2017; Lacroix et al., 2018; Ruffinelli et al., 2020) have shown that KGEs can be effectively trained using cross-entropy loss to predict missing object over $\mathcal{E}$, given subjects and predicates, *i.e.* by maximizing:

$$\log p(o \mid s, r) = \phi_{\text{KGE}}(s, r, o) - \log \sum_{o' \in \mathcal{E}} \exp \phi_{\text{KGE}}(s, r, o') \quad (2)$$

## 6. NTPs as Memory-Augmented Path Algorithm

Inspired by Conditional Theorem Prover (CTP) (Minervini et al., 2020) we can implement NTP as a memory-augmented path-based algorithm. Instead of searching for all rules in the KB, CTP extends NTP by learning a goal transformation module that directly transforms each goal

predicate to a list of predicates following pre-set rule templates (*e.g.* transitivity), thereby forming the proof paths. Given a (sub)goal, the model steps through each atom formed by the transformed goal predicate until it reaches the end of the path. The above procedure is instantiated recursively for each atom (sub-goal) along the path until it reaches the depth limit. This formulation gives us more flexibility for integrating KGE methods comparing to original NTP. In Figure 1 we show a simple example of CTP with $depth = 1$ and one transitive rule template of length $n$. At each step, the process can be viewed as sampling $k$ plausible objects given the subject and predicate $o \sim \mathcal{P}(s, r)$, which shares similar formulation as in formula 2.

## 7. Hardness in Training NTPs

Previous works (Rocktäschel & Riedel, 2017; Maene & De Raedt, 2023; de Jong & Sha, 2019) have primarily focused on analyzing and addressing the limitations of NTPs from the perspective of unsmooth optimization, particularly in relation to the sparse gradient problem. However, attempts to mitigate this issue often introduce additional computational overhead. For example, DeepSoftLog (Maene & De Raedt, 2023) tackles the sparse gradient problem in NTP training by employing differentiable probabilistic semantics, combined with a knowledge compilation step for probabilistic inference, and evaluates the entire proof tree (as opposed to using a top-$k$ approximation) to ensure accurate gradient calculation. While this approach yields improved accuracy and provides a more interpretable probabilistic framework, it struggles to scale beyond small KBs.

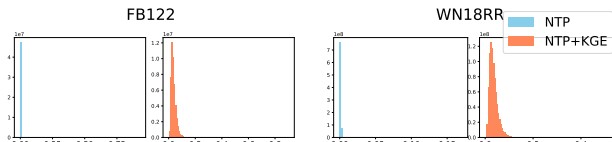

*Figure 2.* Distribution of pairwise similarity from CTP (blue) and CTP combined with KGE (orange).

In contrast to previous works, we try to view the hardness in NTP training from the embedding perspective. Unlike KGEs, which compute triplet scores based on interactions between entities and predicates, NTPs derive embeddings solely from pairwise unification scores. This results in embeddings in NTPs being less structured. Furthermore, while KGEs typically sample a large number of negative examples (*e.g.*, 256) to learn the distribution of entities given a subject/object and relation: $(o \sim \mathcal{P}(s, r))$ or $(s \sim \mathcal{P}(r, o))$, NTPs generally sample only a single negative example per entity and retrieve only the top-$k$ facts from the KB for unification, where $k \ll |\mathcal{E}|$. As a result, semantically similar embeddings in NTPs may end up in vastly different regions of the embedding space if they are never unified or do not

receive gradient updates due to the fuzzy min/max operations. In Figure 2 we show the distribution of pairwise unification scores between entities, and we could observe the pairwise score distribution for CTP (blue) is mostly close to 0, suggesting only a handful of embedded symbols have interactions with each other. This lack of interaction can lead to an unstructured and suboptimal embedding space, negatively affecting the performance of NTPs.

Therefore, given the above challenge in training NTPs, in this work we explore different strategies for leveraging the strengths of KGEs to regularize and enhance the embedding space of NTPs, given the proven effectiveness of KGEs in learning structured representations.

## 8. Method

In this section we discuss the four variants we considered for integrating KGEs with NTPs. In Table 1 we summarize how each variant are implemented on top of CTP.

**KGE as an auxiliary loss model** The most straightforward strategy for leveraging KGEs to support NTP training is to use KGE as an auxiliary model for loss calculation. The overall loss for training NTPs then becomes:

$$\mathcal{L} = (1 - \lambda) \sum_{((s,r,o),y) \, \in \, \mathcal{G}} \Big[ - y \log(\text{KGE}_\theta((s,r,o)) \\ - (1-y) \log(1 - \text{KGE}_\theta((s,r,o))) \Big] + \lambda \mathcal{L}_{\text{NTP}_\theta^\mathcal{G}}$$

where $\lambda$ is a hyper-parameter controlling the weight for the mixture. We denote this variant as $\text{CTP}_1$. Note that using KGE as an auxiliary loss term was briefly mentioned in the original NTP paper (Rocktäschel & Riedel, 2017). However, it was not further examined nor was it used in the subsequent works in GNTP and CTP.

**KGE as an auxiliary score function.** Similar to $\text{CTP}_1$, we again consider utilizing KGE score function. But rather than appending it as a loss term at the very end, here we inject KGE score function $\phi_{\text{KGE}}$ into NTPs as an auxiliary score $\phi_{mixed} = \lambda \phi_{\text{NTP}} + (1 - \lambda) \phi_{\text{KGE}}$. In this way, we could provide additional regularization at each proof step, and force the model to learn interactions between entities and predicates *along* the proof path. We refer to this variant as $\text{CTP}_2$. Despite the simplicity, we find this variant to bring the most consistent improvement across most experiments.

**KGE for stepping through.** For translation-based KGEs such as TRANSE and ROTATE, the tail object $o$ can be efficiently calculated given $(s, r)$. To leverage this translational property, we consider replacing the topk retrieval with a translation-based operation to improve inference efficiency. Specifically, given a $(s, r)$ pair, we use translational KGE to obtain corresponding object and then retrieve the closest

| Modules | CTP | VARIANTS |
|---|---|---|
| step | $i = \text{topk}^{\mathcal{G}}(s,r), k); z = \mathcal{G}[i][-1]$ | CTP$_3$: $\quad z = \text{trans}(s,r); i = \text{topk}^{\mathcal{G}}((s,r,z), k)$ |
| score$_{\text{latent}}$ | $\gamma = \phi_{\text{NTP}}((s,r,z), \mathcal{G}[i])$ | CTP$_2$: $\gamma = \lambda\phi_{\text{NTP}}((s,r,z), \mathcal{G}[i]) + (1-\lambda)\phi_{\text{KGE}}(s,r,z)$ |
| score$_{\text{final}}$ | $i = \text{topk}^{\mathcal{G}}((z,r,o), k); \gamma = \phi_{\text{NTP}}((z,r,o), \mathcal{G}[i])$ | CTP$_4$: $\quad\quad \gamma = \phi_{\text{KGE}}(z,r,o)$ |
| loss | $\mathcal{L} = \mathcal{L}_{\text{NTP}}$ | CTP$_1$: $\quad\quad \mathcal{L} = \lambda\mathcal{L}_{\text{NTP}} + (1-\lambda)\mathcal{L}_{\text{KGE}}$ |

*Table 1.* Summary of the proposed four variants for integration. We consider four modules in CTP to inject KGEs: 1) **step**: Given $(s,r)$ find $o$; 2) **score$_{\text{latent}}$**: unification score along each proof path; 3) **score$_{\text{final}}$**: unification score calculation at the last proof step, and 4) **loss**: the final loss calculation. Column CTP/VARIANT shows the original/modified algorithm by integrating KGEs with the corresponding modules. The variant only differs from the original CTP for the corresponding module (for instance, CTP$_1$ includes the KGE objective in the final loss function. This is the only difference between baseline CTP and CTP$_1$, and all other variants do not include the KGE objective in their loss function). $\mathcal{G}$ denotes the KG, and $\mathcal{G}[i]$ refers to the $i$-th facts in the KG. trans denotes the translation function of KGEs, $z$ refers to the tail entity predicted by $(s,r)$, and $\text{topk}^{\mathcal{G}}$ denotes the top-$k$ retrieval from $\mathcal{G}$ that returns the top-$k$ indices $i$.

$k$ facts for score calculation. During inference we skip the retrieval and score calculation. This variant, referred to as CTP$_3$, is designed to improve the efficiency of NTPs. In this case, for each proof path, CTP$_3$ is very similar to the path-based KGE method PTRANSE (Lin et al., 2015). They differ in (1) PTRANSE follows KGE training strategy, and utilize additional prior for handling spurious relations, while (2) CTP$_3$ calculates unification scores along the proof path, and uses NTP's retrieval-based score calculation.

The replacement of top-$k$ with the 1-1 mapping by translational KGE methods results in two drawbacks: 1. Limited expressiveness due to the top-1 retrieval. 2. Susceptible to spurious proof paths where subject and object are connected but logically irrelevant (Lin et al., 2015). For example, consider the path $(s, born\_in, o_1) \rightarrow (o_1, located\_in, o_2) \rightarrow (o_2, notable\_people, o)$ where $s$ and $o$ are connected but irrelevant. These issues become more obvious when the size and complexity of the dataset increase. Therefore, we further consider two approaches to mitigate these limitations. 1. *Learnable entity expansion module.* we add an additional learnable neural module $\mathcal{W} \in \mathbb{R}^{(d,k)}$ which learns a mapping from the resulting entity embedding $s$ obtained from the translational KGE function to neighboring entities $\{S\}_k$. 2. *Filtering spurious relation paths.* We consider using the Path Constraint Resource Allocation (PCRA) for calculating the path reliability by measuring how much resource flows from the head to the tail entity following a path, as inspired by PTransE (Lin et al., 2015).

**KGE for final score calculation.** We consider applying KGE at the final step at each proof path. One drawback on NTPs' efficiency is their evaluation speed. During evaluation of a link prediction task, in order to rank all the entities in the KG, the model retrieves top-$k$ facts for each combination of the missing entity and the known predicate-object/subject pair, followed by the unification score calculation between two tensors of shape $(|\mathcal{E}|, k, 3d)$ where $k$ is the retrieved $k$ facts, and $d$ is the embedding dimension. For example, WN18RR dataset contains 40,943 entities. With

$k = 10$ we need to compute the pairwise distance between two matrices of shape $(40,943 \times 10, 3d)$. This is done at the end of *every* proof path, leading to extremely slow evaluation compared to KGEs. Therefore, we try to replace the last proof step with KGEs, while keeping the previous steps with NTP. In this way, we wish to leverage the multi-hop reasoning ability of NTPs while using KGEs for local ranking at the final step. We refer to this variant as CTP$_4$.

While it is trivial to combine any variants together, we do not observe noticeable performance gain by doing so. Therefore we leave them separated for clarity.

# 9. Experiments

# 10. Experimental Setups

**Dataset** We conduct experiments on popular link prediction datasets including Countries, Nations, UMLS and Kinship (Kemp et al., 2006). Following GNTP (Minervini et al., 2019) we experiment on FB122 (Guo et al., 2016a), WN18RR (Dettmers et al., 2018), and additionally CoDEx-S (Safavi & Koutra, 2020). FB122 consists of two test splits: Test-I and Test-II, where Test-II contains the set of triplets that can be inferred via logic rules, and Test-I denotes all other triplets. We follow the same evaluation protocol as in GNTP and CTP, and report Mean Reciprocal Rank (MRR) and HITS@$m$ under the filtered setting. Besides, we also test systematic generalization capabilities on CLUTRR (Sinha et al., 2019) dataset, by testing on unseen relations between entities that requires more reasoning steps than the model is trained on, as done in CTP (Minervini et al., 2020).

**Baseline.** We compare our work with NTP-based methods: NTP (Rocktäschel & Riedel, 2017), GNTP (Minervini et al., 2019) and CTP (Minervini et al., 2020), KGE-based COMPLEX (Trouillon et al., 2016) and DIST-MULT (Yang et al., 2015), rule miner such as NEU-RALLP (Yang et al., 2017), MINERVA (Das et al., 2018), DIFFLOGIC (SHENGYUAN et al., 2023),LERP (Han et al., 2023), and GNN NBFNET (Zhu et al., 2021).

| Datasets | Metrics | CTP$_1$ | CTP$_2$ | CTP$_3$ | CTP$_4$ | NTP | GNTP | CTP | NeuralLP | MINERVA | NBFNet | DRUM | LERP |
|---|---|---|---|---|---|---|---|---|---|---|---|---|---|
| Countries S1 | | 99.3 | **100** | 98.1 | 97.8 | 90.8 | **100** | **100** | **100** | **100** | **100** | - | - |
| Countries S2 | AUC-PR | 91.5 | **94.2** | 90.3 | 89.7 | 87.4 | 93.4 | 91.8 | 75.1 | 72.8 | _93.85_ | - | - |
| Countries S3 | | 93.2 | **96.5** | 92.8 | 90.4 | 56.7 | 91.3 | 94.8 | 92.2 | 90.0 | _95.74_ | - | - |
| Kinship | MRR | _0.75_ | 0.71 | 0.5 | 0.59 | 0.35 | 0.72 | 0.71 | 0.62 | 0.72 | **0.80** | 0.53 | 0.64 |
| | HITS@1 | _61.6_ | 57.5 | 49.1 | 48.9 | 24.0 | 58.6 | 56.5 | 47.5 | 60.5 | **63.2** | 36.7 | 50.0 |
| | HITS@3 | _85.0_ | 82.4 | 71.4 | 67.9 | 37.0 | 81.5 | 82.6 | 70.7 | 81.2 | **89.1** | 62.8 | 73.5 |
| | HITS@10 | _96.0_ | 95.6 | 92.8 | 90.1 | 57.0 | 95.8 | 95.3 | 91.2 | 92.4 | **96.6** | 88.5 | 93.1 |
| Nations | MRR | 0.63 | **0.79** | 0.53 | 0.55 | 0.61 | 0.66 | 0.71 | - | - | _0.75_ | - | - |
| | HITS@1 | 44.4 | **68.9** | 31.84 | 34.2 | 45.0 | 49.3 | 56.2 | - | - | _63.3_ | - | - |
| | HITS@3 | 77.6 | **85.6** | 51.92 | 52.8 | 73.0 | 78.1 | 81.3 | - | - | _81.5_ | - | - |
| | HITS@10 | 98.9 | **99.7** | 83.06 | 79.4 | 87.0 | 98.5 | 99.5 | - | - | _95.1_ | - | - |
| UMLS | MRR | 0.82 | **0.85** | 0.65 | 0.70 | 0.80 | _0.84_ | 0.81 | 0.78 | 0.82 | 0.82 | 0.69 | 0.76 |
| | HITS@1 | 69.9 | **75.2** | 54.6 | 62.7 | 70.0 | _73.2_ | 69.4 | 64.3 | 72.8 | 72.1 | 54.6 | 64.6 |
| | HITS@3 | 93.2 | **94.6** | 77.4 | 84.4 | 88.0 | _94.1_ | 89.8 | 86.9 | 90.0 | 89.6 | 80.8 | 85.5 |
| | HITS@10 | 98.7 | **98.2** | 92.6 | 92.2 | 95.0 | _98.6_ | 95.3 | 96.2 | 96.8 | 97.1 | 93.5 | 94.2 |

*Table 2.* Link prediction results on Countries, Kinship, Nations and UMLS datasets. HITS@$m$ are reported as %.

| TransE | RotatE | ComplEx | DistMult | NBFNET | CTP$_2$ |
|---|---|---|---|---|---|
| 9.3 | 40 | 0.32 | 0.24 | 113 | 8.2 |

*Table 3.* Inference speed (millisecond) comparison on WN18RR.

**Implementation.** We conduct our experiments primarily on CTP (Minervini et al., 2020). Since the original CTP did not evaluate on large-scale dataset FB122 and WN18RR, we perform hyper parameter tuning to obtain the CTP baseline. By default, we use COMPLEX for CTP$_1$, CTP$_3$ and CTP$_4$, and ROTATE for CTP$_2$ as we observe best overall performance under these settings. During training, we obtain negative samples by corrupting subject, entity, and both, each with $n$ times, resulting in $3n$ negative samples generated for each triplet. These negative samples will receive negative label $y = 0$, and the model is trained according to the NTP objective (Eq. 1). For CTP$_1$ and CTP$_2$ we use $\lambda = 0.5$ as the default weight for combining KGE and NTP.

## 11. Results

**Link prediction.** In Table 2, 4 and 5 we show link prediction results on the evaluated datasets. In most cases, we can observe CTP$_2$ performs the best among our proposed four variants, and outperforms other NTPs by a large margin, achieving SOTA results on Countries, Nations, UMLS, FB122 and CoDEx-S datasets. While NBFNET achieves best performance on WN18RR, it comes at the cost of efficiency (Table 3), with the longest inference time, $13\times$ more than CTP$_2$. On the other hand, while CTP$_2$ is slower than some KGE-based models, we believe it is still a good compromise between speed and interpretability.

We conjecture the advantage of CTP$_2$ over CTP$_1$ is the injection of KGEs into NTP's chaining process, effectively regularizing each latent subject along the proof path. There-fore, it can be more effective at regularizing the embedding space compared to appending the loss outside the proving process as in CTP$_1$. Moreover, as KGEs are usually trained with large numbers of negatives, directly adding KGE to the loss term of NTP may not be ideal. This can be confirm where CTP$_1$ performs reasonably well on small datasets such as Kinship ($\uparrow 0.04$ MRR compared to CTP$_2$, but significantly lag behind on FB122 ($\downarrow 0.12$ MRR).

**Systematic Generalization.** In Table 6 we show results on CLUTRR (Sinha et al., 2019) datasets, where we train the model on graphs with 2-3 edges and test on graphs with 4-10 edges. We can observe CTP$_2$ has smallest performance degradation as the number of hops increases, with a 0.04 decrease from 4 hops to 10 hops. On the contrary, NBFNet suffers from increasing edge length with an accuracy decrease of over 0.3. In particular, CTP$_2$ shows noticeably stronger performance compared to CTP$_1$ which has little improvements against baseline CTP.

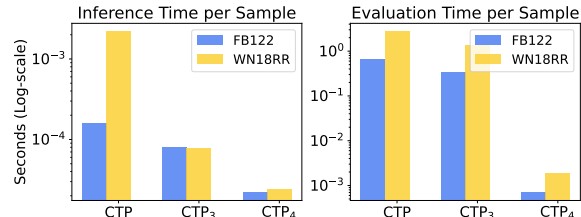

*Figure 3.* Second/sample in **log-scale** on FB122 and WN18RR dataset on a NVIDIA V100 GPU with batch size = 512.

**Boosting NTP speed with KGE** Despite having lower accuracy, CTP$_3$ and CTP$_4$ can significantly improve training and *evaluation* efficiency of NTP, especially on large-scale dataset. In Figure 3, we show per-sample inference and evaluation time under CTP, CTP$_3$ and CTP$_4$. For inference, CTP$_3$ requires $2\times$ and $7\times$ less time compared to CTP on

| | | Test-I | | | | Test-II | | | | Test-ALL | | | |
|---|---|---|---|---|---|---|---|---|---|---|---|---|---|
| | | H@3 | H@5 | H@10 | MRR | H@3 | H@5 | H@10 | MRR | H@3 | H@5 | H@10 | MRR |
| With Rules | KALE$_P$ | **38.4** | **44.7** | **52.2** | 0.32 | 79.7 | 84.1 | 89.6 | 0.68 | 61.2 | 66.4 | 72.8 | 0.52 |
| | KALE$_J$ | 36.3 | 40.30 | 44.90 | **0.33** | 98.0 | 99.0 | 99.2 | 0.948 | 70.7 | 73.1 | 75.2 | 0.67 |
| | ASR$_D$ | 37.3 | 41.0 | 45.9 | 0.33 | **99.2** | **99.30** | **99.4** | **0.984** | 71.7 | 73.6 | 75.7 | 0.67 |
| | KBLRN | - | - | - | - | - | - | - | - | **74.0** | **77.0** | **79.7** | **0.70** |
| Without Rules | TRANSE | 36.0 | 41.5 | 48.1 | 0.29 | 77.5 | 82.8 | 88.4 | 0.63 | 58.9 | 64.20 | 70.2 | 0.48 |
| | DISTMULT | 36.0 | 40.3 | 45.3 | 0.31 | 92.3 | 93.8 | 94.7 | 0.874 | 67.4 | 70.2 | 72.9 | 0.63 |
| | COMPLEX | **37.0** | **41.3** | **46.2** | **0.33** | 91.4 | 91.9 | 92.4 | 0.887 | 67.3 | 69.5 | 0.72 | 0.64 |
| | GNTP | 28.6 | 31.2 | 35.8 | 0.28 | 94.2 | 95.8 | 96.0 | 0.92 | 61.5 | 63.2 | 64.5 | 0.61 |
| | CTP | 31.2 | 34.7 | 39.51 | 0.30 | 96.1 | 97.0 | 97.9 | 0.94 | 64.5 | 65.1 | 68.3 | 0.63 |
| | NBFNet | - | - | - | - | - | - | - | - | 57.2 | 59.6 | 70.7 | 0.51 |
| | CTP$_1$ | 30.6 | 33.1 | 37.8 | 0.29 | 95.0 | 95.9 | 96.6 | 0.89 | 60.4 | 61.3 | 62.9 | 0.56 |
| | CTP$_2$ | 34.4 | 38.2 | 43.1 | 0.32 | **99.1** | **99.2** | **99.4** | **0.98** | **69.9** | **71.32** | **73.0** | **0.68** |
| | CTP$_3$ | 25.3 | 30.2 | 34.2 | 0.25 | 93.7 | 94.5 | 94.8 | 0.83 | 59.4 | 60.8 | 62.2 | 0.53 |
| | CTP$_4$ | 30.2 | 32.7 | 37.1 | 0.28 | 94.5 | 95.4 | 95.9 | 0.85 | 61.1 | 64.6 | 67.4 | 0.61 |

*Table 4.* Link prediction result on FB122 dataset. Following GNTP (Minervini et al., 2019) we report accuracy on Test-I, Test-II and Test-ALL. H@$m$ are reported as %. KALE$_P$ and KALE$_J$ denote KALE-Pre and KALE-Joint from (Guo et al., 2016b). ASR$_D$ denotes ASR-DistMult from (Minervini et al., 2017). All the aforementioned models have access to the ground-truth logic rules.

| | Metrics | CTP$_1$ | CTP$_2$ | CTP$_3$ | CTP$_4$ | GNTP | CTP | COMPLEX | DISTMULT | NEURALLP | MINER. | DRUM | NBFNET | DIFFLOGIC |
|---|---|---|---|---|---|---|---|---|---|---|---|---|---|---|
| WN18RR | MRR | 0.39 | 0.51 | 0.37 | 0.33 | 0.38 | 0.44 | 0.41 | 0.46 | 0.46 | 0.45 | 0.43 | **0.55** | 0.50 |
| | H@1 | 36.4 | 46.2 | 35.2 | 31.8 | 37.1 | 38.6 | 38.2 | 41.0 | 37.6 | 41.3 | - | **49.7** | - |
| | H@3 | 37.9 | 56.6 | 36.6 | 33.5 | 38.5 | 41.2 | 43.3 | 44.1 | 46.8 | 45.6 | - | **57.3** | - |
| | H@10 | 41.5 | 63.5 | 39.4 | 37.7 | 39.5 | 50.7 | 48.0 | 65.7 | 65.7 | 51.3 | 56.5 | **66.6** | 58.7 |
| CoDExS | MRR | 0.30 | **0.48** | 0.35 | 0.27 | 0.29 | 0.32 | 0.40 | 0.44 | - | - | 0.29 | **0.48** | 0.46 |
| | H@1 | 22.5 | 39.5 | 33.4 | 20.5 | 21.4 | 30.2 | 29.3 | 37.2 | - | - | - | **41.9** | - |
| | H@3 | 35.7 | **54.6** | 50.2 | 34.1 | 36.6 | 49.5 | 44.9 | 50.4 | - | - | - | 54.3 | - |
| | H@10 | 50.1 | **66.3** | 58.9 | 49.8 | 52.2 | 56.3 | 62.3 | 64.6 | - | - | 39.5 | 65.5 | 65.5 |

*Table 5.* Link prediction results on WN18RR and CoDEx-S. H@$m$ is reported as %.

| # of Hops | CTP$_1$ | CTP$_2$ | CTP$_L$ | LSTM | MHA | NBFNET |
|---|---|---|---|---|---|---|
| 4 | 0.98 | **0.99** | 0.98 | 0.98 | 0.81 | 0.98 |
| 5 | 0.98 | **0.99** | 0.98 | 0.95 | 0.76 | 0.96 |
| 6 | 0.97 | **0.99** | 0.97 | 0.88 | 0.74 | 0.85 |
| 7 | 0.97 | **0.98** | 0.96 | 0.87 | 0.70 | 0.79 |
| 8 | 0.94 | **0.98** | 0.94 | 0.81 | 0.69 | 0.78 |
| 9 | 0.90 | **0.96** | 0.89 | 0.75 | 0.64 | 0.73 |
| 10 | 0.90 | **0.95** | 0.89 | 0.75 | 0.67 | 0.67 |

*Table 6.* Systematic generalization test on Clutrr, where we train on graphs with 2-3 edges and test on graphs containing 4-10 edges.

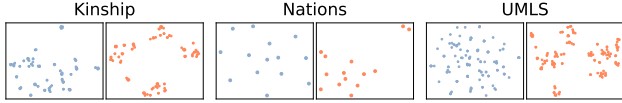

*Figure 4.* $t$-SNE visualization of embeddings for CTP (blue) and CTP$_2$ (orange) with perplexity $= 5$.

FB122 and WN18RR dataset, while CTP$_4$ reduces even further by 28$\times$ and 92$\times$. For evaluation, CTP$_3$ requires 2$\times$ less time than CTP on both datasets, while CTP$_4$ reduces 942$\times$ and 1452$\times$ on FB122 and WN18RR. While performance of CTP$_4$ degrade noticeably, we show in Table 12 (Append.) we can substantially improve its accuracy by incorporating more negative samples during training.

## 12. Ablation Studies

**Regularized Embedding space** In Figure 4 we show the $t$-SNE visualization of the embedding space of original CTP and CTP$_2$-COMPLEX. For both methods, we could observe a few points being close to each other, suggesting the model are able to learn that they are *unifiable*. However, we can clearly observe CTP$_2$-COMPLEX also exhibits better global structures, whereas for CTP there only exists extremely local (pairwise) pattern. On the other hand, as shown in Figure 2, while baseline CTP (left, blue) exhibits extremely sparse connections between entities with unification score all gathered around 0, CTP combined with KGE objectives (right, orange) shows a smoother score distribution, suggesting a much denser connectivity.

In Table 8 we show entity-pairs with their similarity score. At the top of the table we can observe CTP$_2$'s top-10 similar entity-pairs have much higher similarity score, and are more semantically correct as compared to CTP, whose similarity score decreases drastically. At the bottom of Table 8 we show three entity-pairs which are not successfully unified during training of CTP$_2$. We can observe even though they are never unified directly, their similarity scores still show their relevancy as compared to CTP. For instance, the entity pair (*Singer-songwriter*, *Lyricist*) has a score of 0.22 for

| | KGE | UMLS | | | | Kinship | | | | FB122 Test-ALL | | | |
|---|---|---|---|---|---|---|---|---|---|---|---|---|---|
| | | MRR | H@1 | H@3 | H@10 | MRR | H@1 | H@3 | H@10 | MRR | H@3 | H@5 | H@10 |
| CTP | - | 0.80 | 69.4 | 89.8 | 95.3 | 0.70 | 56.56 | 82.64 | 0.95 | 0.63 | 64.5 | 65.1 | 68.3 |
| CTP$_1$ | DistMult | 0.78 | 67.4 | 87.4 | 93.2 | 0.71 | 58.5 | 81.2 | 0.94 | 0.54 | 59.31 | **62.23** | **63.14** |
| | ComplEx | **0.81** | **68.9** | **93.1** | **98.7** | **0.74** | **61.6** | **85.0** | 95.94 | **0.56** | 60.4 | 61.3 | 62.9 |
| | TransE | 0.74 | 61.1 | 83.9 | 96.0 | 0.43 | 32.3 | 47.5 | 64.15 | 0.51 | 57.42 | 60.04 | 62.43 |
| | RotatE | 0.67 | 53.1 | 77.5 | 92.6 | 0.61 | 46.6 | 68.9 | 91.52 | 0.50 | 57.34 | 61.47 | 62.85 |
| CTP$_2$ | DistMult | 0.84 | 74.5 | 93.1 | 98.3 | 0.71 | 59.0 | 79.9 | 93.5 | 0.68 | 69.35 | 72.1 | **73.4** |
| | ComplEx | **0.85** | **75.2** | **94.6** | 98.2 | **0.72** | 57.0 | 82.3 | **95.3** | **0.68** | **69.9** | **71.3** | 73.0 |
| | TransE | 0.83 | 72.1 | 93.3 | 97.0 | 0.71 | 58.7 | **80.6** | 93.9 | 0.64 | 64.1 | 67.4 | 68.2 |
| | RotatE | 0.82 | 70.6 | 93.4 | 98.1 | 0.71 | **59.2** | **80.6** | 93.8 | 0.64 | 65.1 | 68.2 | 69.8 |
| CTP$_3$ | TransE | 0.48 | 36.6 | 57.4 | 78.2 | 0.49 | 40.3 | 68.12 | 90.54 | 0.31 | 28.8 | 35.7 | 44.4 |
| | RotatE | **0.65** | **54.6** | **77.4** | **92.5** | **0.54** | **45.2** | **71.47** | **92.84** | **0.53** | **59.4** | **60.8** | **62.2** |
| CTP$_4$ | DistMult | 0.72 | 57.1 | 78.2 | 89.0 | **0.61** | **49.7** | **69.52** | **91.7** | **0.62** | 62.4 | 64.32 | 66.8 |
| | ComplEx | **0.76** | **62.7** | **84.3** | **92.1** | 0.59 | 48.9 | 67.9 | 90.1 | 0.61 | 61.1 | **64.6** | **67.4** |
| | TransE | 0.58 | 50.3 | 72.4 | 90.1 | 0.53 | 44.6 | 63.21 | 90.5 | 0.48 | 55.3 | 57.8 | 59.0 |
| | RotatE | 0.61 | 49.5 | 74.2 | 91.8 | 0.50 | 43.9 | 63.7 | 89.2 | 0.60 | 61.4 | 64.0 | 67.0 |

*Table 7.* Link prediction results on UMLS, Kinship and FB122 dataset with different KGE models.

| CTP$_2$ | CTP |
|---|---|
| 0.99: (*Baptists, Protestantism*) | 0.92: (*England, UK*) |
| 0.94: (*Christianity, Lutheranism*) | 0.72: (*Musician, Record_producer*) |
| 0.92: (*Christianity, Catholicism*) | 0.59: (*TV_Director, TV_producer*) |
| 0.91: (*English_Lang, French_Lang*) | 0.54: (*Composer, Musician*) |
| 0.90: (*England, United_Kingdom*) | 0.52: (*Composer, Record_producer*) |
| 0.89: (*Lutheranism, Catholicism*) | 0.51: (*Film_Producer, TV_producer*) |
| 0.84: (*Singer, Songwriter*) | 0.40: (*Comedian, TV_producer*) |
| 0.81: (*Singer-songwriter, Musician*) | 0.38: (*Film_Producer, TV_Director*) |
| 0.80: (*Writer, Author*) | 0.38: (*Comedian, Writer*) |
| 0.78: (*Methodism, Catholicism*) | 0.37: (*Will_Smith, Bachelor_of_Arts*) |
| 0.23: (*Los_Angeles, Malibu*) | 0.12: (*Los_Angeles, Malibu*) |
| 0.22: (*Singer-songwriter, Lyricist*) | 0.0003: (*Singer-songwriter, Lyricist*) |
| 0.25: (*Ontario, Canada*) | 0.19: (*Ontario, Canada*) |

*Table 8.* Top: Top-10 entity-pairs ranked by similarity score on FB122. Bottom: entities that are **never** unified during training of CTP$_2$. Green/Red denotes if entity-pair ever been unified or not.

CTP$_2$, but only $3e^{-4}$ for CTP when it is never unified. This suggests CTP$_2$ can interpolate between un-unified entities due to more structured embedding space.

**Effect of weight $\lambda$** We find the weight $\lambda$ for combining NTP and KGE loss/score function plays an important rule on the performance of CTP$_1$ and CTP$_2$. Therefore, we repeat experiments with different $\lambda$ on three tested datasets, with results visualized in Figure 5. We can observe as CTP$_2$ tends to be invariant against $\lambda$, performance of CTP$_1$ on UMLS and Kinship dataset increases when $\lambda$ decreases, yet decrease on FB122 and WN18RR with smaller $\lambda$.

Interestingly, we find that for small datasets such as UMLS and Kinship, CTP$_1$ and CTP$_2$ maintain decent performance when $\lambda = 0$, suggesting both model perform well even when the loss/score is fully substituted by the KGE loss/score. For example, CTP$_1$ achieves SOTA performance of 0.87 MRR on UMLS when $\lambda = 0.1$, and 0.85 with $\lambda = 0$. On the other hand, for bigger datasets such as FB122 and WN18RR

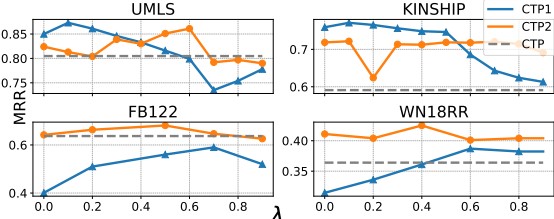

*Figure 5.* MRR with different weight $\lambda$

dataset, CTP$_1$'s performance decreases noticeably when $\lambda = 0$. This suggests the limitation of purely relying on KGE objectives under NTP scheme on larger datasets.

**Effect of using different KGEs** In Table 7 we show the performance of CTP$_1$, CTP$_2$ and CTP$_4$ using different KGE methods: ComplEx, DistMult, TransE and RotatE, and CTP$_3$ with TransE and RotatE. With CTP$_1$ and CTP$_2$, we can observe that the two similarity-based KGEs, ComplEx DistMult generally yields the best performance, whereas translation-based KGE TransE and RotatE often lag back by a large margin. For instance, CTP$_1$ achieves 0.81 MRR on UMLS with ComplEx, but only 0.67 MRR under RotatE. In general, we observe that CTP$_2$ is mostly invariant to the choice of KGE methods, followed by CTP$_1$, whereas CTP$_3$ and CTP$_4$'s performance can vary largely with different KGE methods. This is expected, as CTP$_1$ and CTP$_2$ are using KGE score functions as a regularization term, whereas CTP$_3$ and CTP$_4$ predict directly based on KGEs.

**Improving CTP$_3$** In Table 9, we show results on the effect of Path Constraint Resource Allocation (PCRA) filtering and the Entity Expansion (EE) module for CTP$_3$. We can see both PCRA and EE bring noticeable performance

| | UMLS | | Kinship | | Nations | | FB122 | | WN18RR | | Codex-s | |
| | H@10 | MRR | H@10 | MRR | H@10 | MRR | H@10 | MRR | H@10 | MRR | H@10 | MRR |
|---|---|---|---|---|---|---|---|---|---|---|---|---|
| $CTP_3$ | 92.6 | 0.65 | 92.8 | 0.5 | 83.0 | 0.53 | 62.2 | 0.53 | 39.4 | 0.37 | 58.9 | 0.35 |
| + *PCRA* | 93.3 | 0.66 | 92.7 | 0.50 | 83.1 | 0.54 | 63.6 | 0.55 | 41.6 | 0.39 | 61.7 | 0.38 |
| + *EE* | 94.1 | 0.67 | 93.6 | 0.52 | 83.9 | 0.54 | 65.7 | 0.60 | 43.6 | 0.41 | 64.2 | 0.42 |
| + *Both* | 94.1 | 0.67 | 93.6 | 0.52 | 85.0 | 0.55 | 66.8 | 0.63 | 44.2 | 0.42 | 64.6 | 0.44 |

*Table 9.* Ablations on the Path Constraint Resource Allocation (PCRA) and the learnable Entity Expansion (EE) module for $CTP_3$.

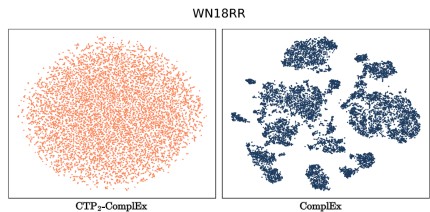

*Figure 6.* $t$-SNE visualization of entity embeddings from trained $CTP_2$-COMPLEX (left) and COMPLEX (right).

gain under most datasets. In particular, both PCRA and EE's performance gains are relatively marginal on smaller KGs such as Kinship and Nations, but are more effective on larer datasets such as FB122 and WN18RR. For example, with both PCRA and EE, $CTP_3$ achieves 0.63 MRR on FB122, a 10% increase over baseline $CTP_3$.

## 13. Limitations

While incorporating KGE objectives into NTPs improves the structural properties of their learned embeddings and enhances performance, certain limitations remain. In Figure 6 we show $t$-SNE visualization of learned embeddings between $CTP_2$ and COMPLEX. Although $CTP_2$ exhibits denser pairwise similarity than the baseline CTP, as shown in Figure 2, its embedding space on large-scale datasets like WN18RR lacks the clear global structure obtained through pure KGE training. We hypothesize two contributing factors: (1) the significantly fewer negative samples used in NTPs compared to KGEs limit the model's ability to capture global structures in large datasets, and (2) the sparse gradient issue becomes more pronounced as dataset size increases. While in this paper we focus on the embedding perspective of NTPs, the sparse gradient problem still remains as a bottleneck to the model performance.

## 14. Conclusion

In this paper we propose to leverage KGE methods to improve NTP performance and efficiency, by enhancing NTP's embedding space to be better structured and regularized, and by replacing computationally expensive NTP components with efficient KGE operations. We explore four variants for integrating KGEs into the NTP, and show that by injecting KGEs into NTP's score calculation ($CTP_2$) we can achieve

the most stable and noticeable improvements across various datasets and configurations. Finally, we conduct detailed ablations and analyze on key components of the integration.

## Impact Statement

This paper presents work whose goal is to advance the field of Machine Learning. There are many potential societal consequences of our work, none which we feel must be specifically highlighted here.

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

## A. Complexity Analysis

Below we provide complexity analysis for the baseline CTP and the proposed variants. We particularly focus on the final step at each proof path during evaluation, since it is the computational bottleneck in the NTP framework.

**Notations:**

Let

- $|\mathcal{E}|$ be the number of entities

- $k$ be the number of top facts retrieved per $(s, r, t)$ pair.

- $d$ be the embedding dimension.

- $N$ be the number of proof path templates.

- $D$ be the depth of the recursive proof tree.

**Baseline CTP** retrieves top-k facts for each combination of the missing entity and the known predicate-object/subject pair, followed by the unification score calculation between two tensors of shape $(|\mathcal{E}|, k, 3d)$. We break down the time complexity for retrieval and score calculation. **Retrieval:** we use the IndexFlatL2 index from FAISS library (Johnson et al., 2019), which has linear complexity $w.r.t.$ number of items. Since we are retrieving for each entity combination, the complexity of retrieval is $O(|\mathcal{E}|^2)$ for the final step at each proof path. **Score calculation:** we employ the Gaussian RBF kernel as the similarity metric between two embeddings. The RBF kernel is defined as

$$\text{RBF}(x_i, y_i) = \exp\left(-\frac{||x_i - y_i||^2}{2\sigma^2}\right),$$

which has approximately linear complexity $w.r.t.$ number of entities $|\mathcal{E}|$ and dimension $d$, $i.e.$ $O(|\mathcal{E}| \cdot kd)$.

Since this has to be done at each individual proof path, the complexity becomes $O(ND(|\mathcal{E}|^2 + |\mathcal{E}| \cdot kd))$.

**CTP$_1$** adds the KGE objective only as a loss term. Therefore the evaluation complexity is the same as baseline CTP$_1$, which is $O(ND(|\mathcal{E}|^2 + |\mathcal{E}| \cdot kd))$.

**CTP$_2$** adds the KGE scoring function on top of baseline CTP's existing similarity function (RBF kernel). The exact additional time complexity depends on the underlying KGE function. However, as we find the additional KGE scoring function is only necessary during training and can be omitted during evaluation time, the evaluation complexity is the same as baseline CTP$_1$, which is $O(ND(|\mathcal{E}|^2 + |\mathcal{E}| \cdot kd))$.

**CTP$_3$** replaces retrievals with translational KGEs to directly compute the next unknown tail entity. This reduces the retrieval complexity from quadratic to linear $w.r.t.$ $|\mathcal{E}|$. Therefore the complexity becomes $O(ND(|\mathcal{E}| + |\mathcal{E}| \cdot kd))$.

**CTP$_4$** replaces the final ranking step from baseline CTP's retrieval + unification score calculation with direct ranking based on the KGE score. Assume a linear complexity for the KGE function $w.r.t.$ number of entities ($e.g.$ ComplEx), the overall complexity is further reduced to $O(ND|\mathcal{E}|)$.

## B. CTP$_1$ vs. CTP$_2$, a gradient perspective

Below we analyze the advantage of CTP$_2$ over CTP$_1$ from a gradient perspective.

Let's first derive the gradient for CTP$_1$ and CTP$_2$.

Recall the loss function for CTP$_1$ is defined as

$$\mathcal{L}_{\text{CTP}_1} = (1 - \lambda) \sum_{((s,r,o),y) \in \mathcal{G}} \left[ -y \log(\text{KGE}_\theta((s, r, o)) - (1 - y) \log(1 - \text{KGE}_\theta((s, r, o)) \right] + \lambda \mathcal{L}_{\text{NTP}_\theta^\mathcal{G}} \tag{3}$$

To simplify notation, we omit $\lambda$ and summation in the following derivations. Let $\theta$ denotes the learnable embeddings, $\phi_N$ and $\phi_K$ denote the score function of KGE and the similarity metic of NTP, we can rewrite Eq. 3 as

$$\mathcal{L}_{\mathrm{CTP}_1}(\theta) = -y \log \phi_N(\theta) - (1-y) \log(1 - \phi_N(\theta)) - y \log \phi_K(\theta) - (1-y) \log(1 - \phi_K(\theta)).$$

Applying the chain rule, the gradient of $\mathcal{L}_{\mathrm{CTP}_1}$ with respect to $\theta$ is

$$
\begin{aligned}
\nabla_\theta \mathcal{L}_{\mathrm{CTP}_1} &= \left[ -\frac{y}{\phi_N(\theta)} + \frac{1-y}{1 - \phi_N(\theta)} \right] \nabla_\theta \phi_N(\theta) + \left[ -\frac{y}{\phi_K(\theta)} + \frac{1-y}{1 - \phi_K(\theta)} \right] \nabla_\theta \phi_K(\theta) \\
&= -\frac{y}{\phi_N(\theta)} \nabla_\theta \phi_N(\theta) + \frac{1-y}{1 - \phi_N(\theta)} \nabla_\theta \phi_N(\theta) + (-\frac{y}{\phi_K(\theta)}) \nabla_\theta \phi_K(\theta) + \frac{1-y}{1 - \phi_K(\theta)} \nabla_\theta \phi_K(\theta)
\end{aligned}
\tag{4}
$$

For CTP$_2$, the score function is a combination (assume $\lambda = \frac{1}{2}$) of KGE and original NTP's similarity score, write as

$$\phi_{\mathrm{comb}}(\theta) = \frac{\phi_N(\theta) + \phi_K(\theta)}{2},$$

and the loss becomes

$$\mathcal{L}_{\mathrm{CTP}_2}(\theta) = -y \log \phi_{\mathrm{comb}}(\theta) - (1-y) \log(1 - \phi_{\mathrm{comb}}(\theta)).$$

By the chain rule, its gradient is

$$\nabla_\theta \mathcal{L}_{\mathrm{CTP}_2} = \frac{d\mathcal{L}(\phi_{\mathrm{comb}})}{d\phi_{\mathrm{comb}}} \nabla_\theta \phi_{\mathrm{comb}}(\theta),$$

where

$$\frac{d\mathcal{L}(\phi_{\mathrm{comb}})}{d\phi_{\mathrm{comb}}} = -\frac{y}{\phi_{\mathrm{comb}}} + \frac{1-y}{1 - \phi_{\mathrm{comb}}}.$$

The gradient for $\phi_{\mathrm{comb}}(\theta)$ is

$$\nabla_\theta \phi_{\mathrm{comb}}(\theta) = \nabla_\theta \phi_N(\theta) + \nabla_\theta \phi_K(\theta),$$

therefore, the gradient for each individual score calculation in CTP$_2$ is

$$
\begin{aligned}
\nabla_\theta \mathcal{L}_{\mathrm{CTP}_2} &= \frac{d\mathcal{L}(\phi_{\mathrm{comb}})}{d\phi_{\mathrm{comb}}} \cdot \nabla_\theta \phi_{\mathrm{comb}}(\theta) \\
&= \left[ -\frac{y}{\phi_{\mathrm{comb}}(\theta)} + \frac{1-y}{1 - \phi_{\mathrm{comb}}(\theta)} \right] \cdot (\nabla_\theta \phi_N(\theta) + \nabla_\theta \phi_K(\theta)) \\
&= -\frac{y}{\phi_{\mathrm{comb}}(\theta)} \cdot \nabla_\theta \phi_N(\theta) + (-\frac{y}{\phi_{\mathrm{comb}}(\theta)}) \cdot \nabla_\theta \phi_K(\theta) + \frac{1-y}{1 - \phi_{\mathrm{comb}}(\theta)} \cdot \nabla_\theta \phi_N(\theta) + \frac{1-y}{1 - \phi_{\mathrm{comb}}(\theta)} \cdot \nabla_\theta \phi_K(\theta).
\end{aligned}
\tag{5}
$$

By comparing the gradient for CTP$_1$ (Eq. 4) and CTP$_2$ (Eq. 5), we can see that CTP$_1$'s gradient only involve 'self' terms, e.g. $-\frac{y}{\phi_N(\theta)} \nabla_\theta \phi_N(\theta)$, where the gradients of KGE and NTP are separately computed and then added. On the other hand, CTP$_2$'s gradient involves 'cross' terms, such as $-\frac{y}{\phi_{\mathrm{comb}}(\theta)} \cdot \nabla_\theta \phi_N(\theta)$ and $(-\frac{y}{\phi_{\mathrm{comb}}(\theta)}) \cdot \nabla_\theta \phi_K(\theta)$, which could potentially provide a more coherent gradient update.

We can further consider the above implication from a gradient variance perspective. Assume both $\phi_N$ and $\phi_K$ are noisy estimates of an underlying true signal

$$\phi_N(\theta) = \phi + \epsilon_N, \quad \phi_K(\theta) = \phi + \epsilon_K,$$

with independent zero-mean terms $\epsilon_N, \epsilon_K$ each with variance $\sigma^2$. Assume further

$$\nabla_\theta \phi_N(\theta) \approx \nabla_\theta \phi_K(\theta) \equiv \nabla_\theta \phi.$$

Define

$$g(\phi) = -\frac{y}{\phi} + \frac{1-y}{1-\phi} \quad \text{and} \quad g'(\phi) = \frac{y}{\phi^2} + \frac{1-y}{(1-\phi)^2}.$$

Using a first-order Taylor expansion, we can approximate

$$g\big(\phi_N(\theta)\big) \approx g(\phi) + g'(\phi)\epsilon_N, \quad g\big(\phi_K(\theta)\big) \approx g(\phi) + g'(\phi)\epsilon_K.$$

The gradient for $\text{CTP}_1$ becomes:

$$\nabla_\theta L_{\text{CTP}_1} \approx \Big[g(\phi) + g'(\phi)\epsilon_N\Big]\nabla_\theta\phi + \Big[g(\phi) + g'(\phi)\epsilon_K\Big]\nabla_\theta\phi$$
$$= 2g(\phi)\nabla_\theta\phi + g'(\phi)(\epsilon_N + \epsilon_K)\nabla_\theta\phi.$$

The noise term is

$$\Delta_{\text{CTP}_1} = g'(\phi)(\epsilon_N + \epsilon_K)\nabla_\theta\phi,$$

with variance

$$\text{Var}[\Delta_{\text{CTP}_1}] \propto (g'(\phi))^2 \text{Var}(\epsilon_N + \epsilon_K) = (g'(\phi))^2 \cdot 2\sigma^2. \tag{6}$$

For $\text{CTP}_2$, the combined score is

$$\phi_{\text{comb}} = \phi + \frac{\epsilon_N + \epsilon_K}{2}.$$

Linearizing,

$$g\big(\phi_{\text{comb}}\big) \approx g(\phi) + g'(\phi)\frac{\epsilon_N + \epsilon_K}{2}.$$

Thus, the gradient for $\text{CTP}_2$ becomes

$$\nabla_\theta L_{\text{CTP}_2} \approx \left[g(\phi) + g'(\phi)\frac{\epsilon_N + \epsilon_K}{2}\right]\nabla_\theta\phi.$$

The noise term is

$$\Delta_{\text{CTP}_2} = g'(\phi)\frac{\epsilon_N + \epsilon_K}{2}\nabla_\theta\phi,$$

with variance

$$\text{Var}[\Delta_{\text{CTP}_2}] \propto (g'(\phi))^2 \text{Var}\left(\frac{\epsilon_N + \epsilon_K}{2}\right) = (g'(\phi))^2 \frac{1}{4} \cdot 2\sigma^2 = (g'(\phi))^2 \frac{\sigma^2}{2}. \tag{7}$$

By comparing $\text{Var}[\Delta_{\text{CTP}_1}]$ and $\text{Var}[\Delta_{\text{CTP}_2}]$ (Eq. 6 and Eq. 7), we can see the variance of the gradient noise for $\text{CTP}_2$ is reduced by a factor of 4 $w.r.t.$ $\text{CTP}_1$'s. This reduction could also lead to smoother and more stable optimization dynamic, which is also what we observed empirically.

## C. Dataset Information

We conduct experiments on three small-scale link prediction datasets: Kinship, Nations and UMLS (Kemp et al., 2006), as well as two large-scale Knowledge Graph (KG) datasets: FB122 (Guo et al., 2016a) and WN18RR (Dettmers et al., 2018). FB122 is a subset of Freebase (Bollacker et al., 2007) containing facts of people, location and sports. Its test set is splitted into two subsets, Test-I and Test-II, where Test-I contains all triplets that *cannot* be derived by deductive logic inference, and Test-II denotes all the rest triplets. WN18RR is derived from WordNet (WN18) (Miller, 1995), where test triplets that can be obtained by inverting triplets in the training set are removed. In Table 10 we summarize the statistics of these datasets.

| Dataset | $|\mathcal{E}|$ | $|\mathcal{R}|$ | # Train | # Validation | # Test |
|---|---|---|---|---|---|
| Kinship (Kemp et al., 2006) | 104 | 25 | 8544 | 1068 | 1074 |
| Nations (Kemp et al., 2006) | 14 | 55 | 1592 | 199 | 201 |
| UMLS (Kemp et al., 2006) | 135 | 46 | 5,216 | 652 | 661 |
| FB122 (Guo et al., 2016a) | 9738 | 122 | 91,638 | 9595 | 11243 |
| WN18RR (Dettmers et al., 2018) | 40,943 | 11 | 86,835 | 3,034 | 3,134 |

*Table 10.* **Dataset statistics** Statistics of datasets used in this work. Columns: number of entities ($|\mathcal{E}|$), number of predicates ($|\mathcal{R}|$), number of training, validation, and test samples.

| Kinship | Nations | UMLS | FB122 | WN18RR |
|---|---|---|---|---|
| *term21*(X,Y):- *term24*(Y,Z) *term4*(X,Y):- *term4*(Y,Z) *term9*(X,Y):- *term11*(Y,Z) | *treaties*(X,Y):- *treaties*(Y,X) *aidenemy*(X,Y):- *militaryactions*(Y,X) *lostterritory*(X,Y):- *timesincewar*(Y,X) | *associated_with*(X,Y):- *process_of*(X,Y), *process_of*(Y,Z) *occurs_in*(X,Y):- *issue_in*(X,Y), *process_of*(Y,Z) *interconnects*(X,Y):- *result_of*(X,Y), *result_of*(Y,Z) | *contains*(X,Y) :- *capital*(Y, X) *language_spoken*(X, Y):- *official_language*(X,Y) *place_lived*(X,Y):- *place_of_birth*(X,Y) | *hypernym*(X,Y):- *hypernym*(Y,X) *verb_group*(X,Y):- *verb_group*(Y,X) *has_part*(X,Y) :- *part_of*(Y,X) |

*Table 11.* Visualization of learnt rules under each dataset with $\text{CTP}_2$

# D. Experimental Settings

**Indexing Library.** In this work we use the FAISS search index (Johnson et al., 2019). We use the GPU version of the library and use the IndexFlatL2 index, which performs exact search using L2.

**Rule templates** CTP defines a number of rule templates for the model to explore. The template is defined as number of steps – how many steps to hop from the head to the tail entity, and whether it is a reverse relation, indicated by $r$, *i.e.* stepping from tail to head entity. For example, rule $= 0$ means the model will try to directly unify the goal with facts in the KB. rule $= 2$ means two steps from the head to the tail entity, *e.g.* $R(s,o) :- R_1(s,z), R_2(z,o)$. rule $= 1R$ means a reverse relation: $R(s,o) :- R_1(o,s)$. For Kinship, Nations and UMLS we follow the setting in CTP with Kinship=$\{0, 1, 1r\}$, Nations=$\{0, 2, 1r\}$, and UMLS=$\{0, 2\}$. For FB122 and WN18RR we both use $\{0, 1, 2, 1r\}$.

**Training** For hyper-parameters we follow CTP (Minervini et al., 2020) on Kinship and UMLS datasets for all the experiments. Specifically, we use embedding_size=50, top-$k$=4, batch_size=8, learning_rate=0.1, trained 100 epochs with Adagrad optimizer. For each triplets we sample 3 negative sample per entity (a total of 9 negative samples per triplet). For Nations we use batch_size=256 with AdamW optimizer for the $\text{CTP}_2$ variant, and the same as CTP for the rest of models. For FB122 we mostly follow the setting from GNTP (Minervini et al., 2019), with embedding_size=100, top-$k$=10, and 1 negative sample per entity. We use Adagrad optimizer and train 100 epochs. We find it necessary to keep a large number of retrieval (*e.g.* 128) for larger datasets such as WN18RR. To avoid GPU OOM during evaluation due to the large matrix operation, we only scale number of retrievals during training, while keeping the number of retrievals during inference or evaluation low.

For baseline CTP we find freezing the model entities in the first 25 epochs work well, while for all our CTP variants we receive better results by not freezing the model from the beginning. We also explore different score aggregation operations for aggregating scores along one proof path (AND operation). For baseline CTP and $\text{CTP}_1$ we find the original min generally work well, while mean and multiplication work better for $\text{CTP}_2$, $\text{CTP}_3$ and $\text{CTP}_4$. besides, we considered using cosine similarity as the scoring metric, with using addition instead of concatenation for obtaining the embedding for the whole triplets. However, we do not observe it to perform better than using the Gaussian kernel.

**Incorporating KGE objectives** To ensure KGE score lies within 0 and 1 we add a Sigmoid function to its negative score function. To avoid small negative scores being pushed to zeros after Sigmoid, we first subtract the mean from the negative scores.

| $n$ | Metrics | CTP | CTP$_1$ | | | CTP$_2$ | | | CTP$_3$ | CTP$_4$ |
|---|---|---|---|---|---|---|---|---|---|---|
| | | | $\lambda=0$ | $\lambda=0.5$ | $\lambda=0.8$ | $\lambda=0$ | $\lambda=0.5$ | $\lambda=0.8$ | | |
| 1 | MRR | **0.64** | 0.40 | 0.56 | **0.59** | 0.64 | **0.68** | 0.65 | 0.53 | 0.32 |
| | HIT@3 | **64.50** | 46.24 | 60.40 | **62.43** | 0.65 | **69.43** | 65.83 | 59.40 | 34.80 |
| | HIT@10 | **68.30** | 50.07 | 62.90 | **63.75** | 67.50 | **73.01** | 68.76 | 62.20 | 41.52 |
| 16 | MRR | 0.61 | 0.43 | **0.59** | 0.57 | 0.63 | 0.65 | 0.49 | **0.55** | 0.45 |
| | HIT@3 | 62.50 | 47.62 | **60.17** | 59.49 | 64.25 | 67.03 | 50.03 | **60.84** | 48.90 |
| | HIT@10 | 65.71 | 51.43 | **62.49** | 61.84 | 65.79 | 69.25 | 53.81 | 61.53 | 41.09 |
| 32 | MRR | 0.56 | **0.43** | 0.58 | 0.48 | 0.63 | 0.62 | 0.46 | 0.54 | 0.59 |
| | HIT@3 | 57.26 | **50.86** | 58.94 | 49.72 | 64.70 | 64.50 | 49.31 | 59.58 | 60.46 |
| | HIT@10 | 59.94 | **53.67** | 60.12 | 51.88 | 66.02 | 67.62 | 53.18 | 63.84 | 62.62 |
| 128 | MRR | - | - | - | - | - | - | - | - | **0.61** |
| | HIT@3 | - | - | - | - | - | - | - | - | **61.10** |
| | HIT@10 | - | - | - | - | - | - | - | - | **67.40** |

*Table 12.* Test results on FB122 with different number of negative samples $n$. Due to the computational limit, we only evaluate CTP$_4$ when $n = 128$. Bold denotes column-wise best results.

# E. Training Dynamics

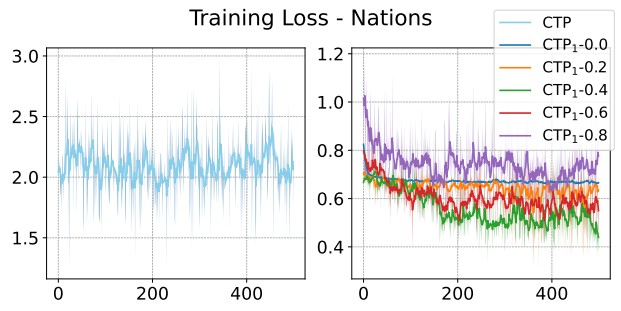

*Figure 7.* Training loss on Nations with CTP (left, blue) and CTP$_2$-COMPLEX (right) with different $\lambda$.

*Figure 8.* Validation MRR on FB122 dataset with baseline CTP and CTP$_2$ with DISTMULT and COMPLEX as integrated KGEs.

In Figure 7 we show training loss on Nations dataset given different $\lambda$. For CTP$_1$ we find the training loss tends to be much more stable with smaller $\lambda$ as shown in Figure 2, which is as expected as the non-differentiable operations in CTP is smoothed out by the differentiable KGE loss calculation. On the other hand, the training loss for baseline CTP is always fluctuating without decreasing, even though the final test accuracy is reasonable.

In Figure 8 we show validation MRR during training on FB122 dataset for baseline CTP (blue), CTP$_2$ with COMPLEX (green) and DISTMULT (orange). We can observe that both CTP$_2$ converges quickly in the first 20 epochs, with CTP$_2$-COMPLEX slightly higher than CTP$_2$-DISTMULT, and both have much higher accuracy than the baseline CTP.

# F. Visualization of learnt rules

In Table 11 we show visualization results generated under each dataset under CTP2. We can see it successfully learns logical rules such as *place_lived*(X,Y):- *place_of_birth*(X,Y), *interconnects*(X,Y):- *result_of*(X,Y), *result_of*(Y,Z), and *contains*(X,Y) :- *capital*(Y, X).

# G. Training NTPs with more negatives

Since KGEs, under the energy-based-model interpretation, require large amount of negative samples, we want to see if such can benefit the integration of NTPs and KGEs, given NTPs' number of negative samples is default to 1. In Table 12 we show results where we increase number of negative samples during training our CTP variants. Interestingly, instead of

receiving better accuracy, we observe a drastic performance drop on CTP, $CTP_1$ when $\lambda = 0.8$ and $CTP_2$ with $\lambda = 0.5$ and $\lambda = 0.8$. For example, the MRR of $CTP_1$ with $\lambda = 0.8$ drops from 0.59 to 0.48 when number of negatives is increased from 1 to 32, and the MRR for $CTP_2$ with $\lambda = 0.8$ drops from 0.65 to 0.46. Reversely, when $\lambda = 0$, $CTP_1$'s MRR increases from 0.40 to 0.438 as number of negatives increases. This implies increasing the number of negatives helps when $\lambda$ is low, $i.e.$ when the KGE loss is contributing more to the gradient updates. However, even when $\lambda = 0$ for $CTP_1$, recovering a pure KGE optimization process, the accuracy with $n = 32$ is still far less than when $\lambda = 0.5$ and all other variants. This suggests that the regularization of KGEs on NTPs is still limited, and we conjecture the bottleneck is the sparse gradient problem introduced during training. As we do not focus on sparse gradient problem in this work, we will leave it for future exploration. On the other hand, we notice drastic increase in accuracy with $CTP_4$ from 0.32 MRR to 0.61 with $n$ increases from 1 to 128.

## H. Details for Improving $CTP_3$'s Performance

In this section we discuss why $CTP_3$'s performance could be drastically lagging behind other variants especially on large datasets, and propose two approaches to improve its performance. Recall for $CTP_3$ we replace top-$k$ retrieval by using translational KGEs ($e.g.$ TransE) to directly compute the unknown entity, which means we are effectively doing top-1 retrieval. This results in two main drawbacks: 1. Limited expressiveness due to the top-1 retrieval. 2. Susceptible to spurious proof paths where subject and object are connected but logically irrelevant. For example, consider the path $(s, born\_in, o_1)$, $(o_1, located\_in, o_2)$, $(o_2, notable\_people, o)$, where $s$ and $o$ are connected but irrelevant. These issues become more obvious when the size and complexity of the dataset increase.

To mitigate the above issues, we additionally propose two methods.

**Filtering spurious relation paths.** We consider using the Path Constraint Resource Allocation (PCRA) for calculating the path reliability by measuring how much resource flows from the head to the tail entity following a path, as inspired by PTransE (Lin et al., 2015).

Formally, for a pair of $s$ and $o$ and a path $p = (r_1, r_2, \ldots, r_n)$, the flow path can be written as $s \xrightarrow{r_1} S_1 \xrightarrow{r_2} \ldots \xrightarrow{r_n} S_n$, where $S_i$ are sets of entities, and $o \in S_n$. Following the notation in PTransE, given any entity $m \in S_i$, the resource flowing to $m$ is defined as

$$R_p(m) = \sum_{n \in S_{i-1}(\cdot, m)} \frac{1}{|S_i(n, \cdot)|} R_p(n),$$

where $S_{i-1}(\cdot, m)$ are its direct predecessors in $S_{i-1}$, and $S_i(n, \cdot)$ is the direct successors of $n \in S_{i-1}$. By calculating $R_p(m)$ recursively from $s$ to $o$, we can obtain the final resource (reliability) of the path $p$ given $s$ and $o$. For more details please refer to (Lin et al., 2015). During training, we then mask out the path with lowest 10% reliability score (we do not modify for evaluation stage).

**Learnable entity expansion module.** To alleviate the issue with top-1 retrieval, we consider adding an additional learnable module which expands the resulting entity embedding obtained from the translational KGE function to $k$ neighboring entities. In other words, we learn a linear layer $\mathbf{W}$ with shape $(d, k)$ to encode a set of related entities $\{S_i\}_k$ given the calculated entity $s$, $i.e.$ $p(\{S_i\}_k|s, \mathbf{W})$.

In Table 9 we show the results for incorporating PCRA and entity expansion module for $CTP_3$. We can observe noticeable improvements, especially on larger datasets such as WN18RR.

## I. Settings for Inference speed Comparison (Table 3)

We run inference on the selected models on a V100 GPU with a batch size of 8, embedding dimension 1000 for each KGE method (default), and $CTP_2$ with dimension 100, and NBFNet with dimension=32.

# J. Pseudo-code implementation

**Neural Theorem Prover** implements backward chaining algorithm by recursively instantiating AND/OR modules, where OR is called to prove each goal by unifying with each rule head in the KB. Then, the AND module is called to prove the rule body, where for each atom in the body the OR is recursively called, until the algorithm reaches depth limit $d$. The pseudo-code for NTP can be found in 1.

# K. Conditional Theorem Prover

**Conditional Theorem Prover** extends upon NTP by incorporating a trainable neural module for predicting plausible rules given goals. The pseudo-code for CTP can be found in 2.

---

**Algorithm 1** Python pseudo-code for NTP with top-$k$ retrieval following implementation from (Minervini et al., 2019)

```python
# KB: the Knowledge Base.

# S: proof state
#  - score: unification score
#  - subs: substitution set

# sim: similarity function for unification
# topk: a function that performs top-k retrieval

def or(goal, S, k):

    S_list = []
    for rule in KB:
        head, body = rule
        topk_ind = topk(goal, KB)
        if d < max_depth and no_cycle(S.subs, rule):
            S_head = unify(head, goal, S, topk_ind)
            S_head = kmax(goal, S_head)
            S_body = and(body, S, d)
            S_list.append(S_body)

    return proof_states

def and(goal, S):

    S_list = []
    if len(goal) == 0:
        S_list = [S]
    elif d < max_depth:
        goal, sub_goals = goal
        new_goal = substitute(goal, subs)
        for S_new in or(new_goal, S, d+1):
            S_list.append(and(sub_goals, S_new))

    return S_list

def unify(atom, goal, S, topk_ind):

    grounded_atom, grounded_goal = [], []
    for (atom_term, goal_term) in zip(atom, goal):

        if is_variable(atom_term):
            if atom_term not in S.subs:
                S.subs.update({atom_term: goal_term}
        elif is_variable(goal_term):
            if is_grounded(atom_term) and \
                goal_term not in S.subs:
                S.subs.update({goal_term: atom_term}

        elif is_grounded(atom_term) and \
                is_grounded(goal_term):
            grounded_atom.append(atom_term)
            grounded_goal.append(goal_term)

            score = sim(grounded_goal, \
                grounded_atom[topk_ind])
            S.score = min(S.score, score)

    return S
```

---

**Algorithm 2** Simplified Python pseudo-code for CTP following (Minervini et al., 2020)

```python
# KB: the Knowledge Base.
# sim: similarity function for unification
# topk: a function that performs top-k retrieval
# max_depth: maximum recursive depth

def ctp(s, r, o, max_depth):

    if max_depth == 0:
      return unify(s, r, o)
    else:
      score = None
      for d in range(max_depth):

        level_score = None
        for rule_path in rule_templates:

          path_score = None
          for step_ind, rule_transform in rule_path:

            if is_inverse_relation:
              latent_score, s = \
                    step(o, r, s, max_depth - 1)
            else:
              latent_score, s = \
                    step(s, r, o, max_depth - 1)

            if path_score is None:
              path_score = latent_score
            else:
              # min aggregation --
              # all proofs need to be hold.
              path_score = min(path_score, latent_score)

            if step_ind == len(rule_path):
              # choose the max over the topk branches
              path_score, _ = max(path_score, dim=-1)

          if level_score is None:
            level_score = path_score
          else:
            # max aggregation --
            # only one proof path needs to be hold.
            level_score = max(level_score, path_score)

        if score is None:
          score = level_score
        else:
          # max aggregation --
          # only one proof path needs to be hold.
          score = max(score, level_score)

def unify(s,r,o=None):

    if o is not None:
        topk_ind = topk([s, r, o])
    else:
        topk_ind = topk([s, r])
        o = KB[topk_ind][-1]

    score = sim([s, r, o], KB[topk_ind])
    return score, o
```

