# OpenReview forum: "Improving Soft Unification with Knowledge Graph Embedding Methods"
_ICML.cc/2025/Conference — ICML 2025 poster_

### Official Review · Reviewer_ck5p · 2025-03-12

**Overall Recommendation:** 3

**Summary:**

This paper presents the first integration of NTP and KGE, aiming to enhance the performance of NTP in terms of both effectiveness and efficiency. The experimental results demonstrate the synergy between these two lines of research.

**Claims And Evidence:**

To some extent, yes. There are a few claims in the introduction that are not entirely accurate, which I have outlined in the 'Other Weaknesses' section below.

**Essential References Not Discussed:**

### Inductive KGE/KGFMs on unseen relations
[1]. Galkin, M., Yuan, X., Mostafa, H., Tang, J., and Zhu, Z. Towards foundation models for knowledge graph reasoning. In ICLR, 2024.

[2]. Mao, H., Chen, Z., Tang, W., Zhao, J., Ma, Y., Zhao, T., Shah, N., Galkin, M., and Tang, J. Position: Graph foundation models are already here. In ICML, 2024.

[3]. Lee, J., Chung, C., and Whang, J. J. Ingram: Inductive knowledge graph embedding via relation graphs. In ICML, 2023.

### Inductive GNNs on unseen nodes (there are more)
[4]. Waikhom, Lilapati, Yeshwant Singh, and Ripon Patgiri. "PO-GNN: Position-observant inductive graph neural networks for position-based prediction." Information Processing & Management 60.3 (2023)

**Experimental Designs Or Analyses:**

Yes, I believe the authors have provided sufficient details about the experiments, which are undoubtedly comprehensive. Additional comments can be found in the 'More Strengths' section.

**Methods And Evaluation Criteria:**

Yes.

**Other Comments Or Suggestions:**

## Weaker suggestions
1. More explanation of the 'sparse gradient perspective' in the Introduction is needed before shifting the focus to the embedding space.

**Other Strengths And Weaknesses:**

## Strengths
1. The paper is well-structured overall, and the integration of the two research areas is clearly demonstrated.
2. The experimental section is thorough, evaluating four potential methods for integrating KGE, while also comparing different KGE variants. The results show an overall improvement when integrating KGE, aligning with the authors' claims. Additionally, the increase in inference time is presented.

## Weaknesses
1. In the Introduction, it is initially stated that DL is limited by poor interpretability. However, later it is suggested that the transformation of discrete symbols into continuous vector space combines the strengths of both DL and symbolic AI. I wonder whether LLMs also fall under the category of NeSy AI, yet they still struggle with explanation abilities. Moreover, theorem proofs require significantly more interpretability than natural language, which raises the question of whether this approach is suitable for such domains.
2. The connection between NTP and KGE could be better introduced. Additionally, there are inductive KGE methods and GNNs that handle unseen entities or relations by defining local representations for unseen items. Knowledge graph foundation models also address such generalization issues [1-4].
3. It would be valuable to explore the potential reasons behind the observed improvements and identify which types of proof structures could benefit most from KGE integration. This would provide further insight to the research community.

**Questions For Authors:**

N/A

**Relation To Broader Scientific Literature:**

The integration of KGE to improve NTP is directly beneficial for scientific discovery, as NTP has become a key research focus for many logicians and mathematicians, particularly in areas like automated proof checking and broader scientific applications.

**Theoretical Claims:**

N/A

---

> ### Author Rebuttal · Authors · 2025-03-30
>
> > ***1. In the Introduction ... raises the question of whether this approach is suitable for such domains.***
>
> Thank you for your insights! While LLMs also transform discrete symbols to continuous space, the difference is NeSy approaches incorporate more explicit reasoning priors in their framework (e.g. backward chaining algorithm for NTPs), as compared to the generic next-token-prediction in LLMs. These structural priors allow NeSy models to have less hallucination and better interpretability (as one can trace the proof paths and localize errors).
> > ***2. The connection between NTP and KGE could be better introduced / include works of inductive KGE/KGFMs on unseen relations***
>
> We will revise our manuscript to incorporate more corresponding details. Also, just to clarify, in this work we do not focus on inductive KGC, and hence we did not include these works initially.
>
> > ***More explanation of the 'sparse gradient perspective'***
>
> We will rewrite the introduction to add more explanations.
> > ***3. potential reasons behind the observed improvements***
>
> **Improved Embedding space.** First, we empirically find integrating KGE (paricularly CTP2) can notably improve the structureness of NTP embedding space. This is discussed throughout the paper.
>
> **Explanation of why similarity-based KGEs work better with NTPs than translational-based KGEs.** Please refer to section *2.why using similarity-based KGEs can achieve better performance* in our reply to Reviewer Bi1c.
>
> **Theoretical justification CTP1 vs. CTP2.** We provide a theoretical justification from a gradient perspective of why CTP2 performs better than CTP1 (as CTP2 is different from CTP3 and CTP4 and mostly similar to CTP1). Due to space limit, we are only able to show essential steps.
>
> Let $\theta$ denotes the learnable embeddings, $\phi_N$ and $\phi_K$ denote the score function of KGE and CTP, the loss for CTP1 is
> $$\mathcal{L_1}(\theta) = -y \log \phi_N(\theta) - (1-y) \log(1-\phi_N(\theta)) - y \log \phi_K(\theta) - (1-y) \log (1-\phi_K(\theta)).$$
> Applying the chain rule, the gradient of $\mathcal{L_1} \ \text{with respect to} \  \theta$  is
> $$\nabla_\theta \mathcal{L_1} = \left[-\frac{y}{\phi_N(\theta)} + \frac{1-y}{1-\phi_N(\theta)}\right] \nabla_\theta \phi_N(\theta) + \left[-\frac{y}{\phi_K(\theta)} + \frac{1-y}{1-\phi_K(\theta)}\right] \nabla_\theta \phi_K(\theta).\quad(1)$$
> For CTP2, the score function is a combination (assume $\lambda = \frac{1}{2}$) of KGE and original NTP's similarity score, write as
> $$\phi_2(\theta) = \frac{\phi_N(\theta) +  \phi_K(\theta)}{2}, $$
> and the loss becomes
> $$\mathcal{L_2}(\theta) = -y\log\phi_2(\theta)-(1-y)\log(1 -\phi_2(\theta)).$$
> the gradient for each individual score calculation in CTP2 is
> $$\nabla_\theta \mathcal{L_2}=\left[-\frac{y}{\phi_2(\theta)} + \frac{1-y}{1-\phi_2(\theta)}\right] \cdot(\nabla_\theta \phi_N(\theta) + \nabla_\theta \phi_K(\theta))\quad(2)$$
> By comparing Eq.1 and Eq.2 we can see that CTP1's gradient only involve "self" terms, where the gradients of KGE and NTP are separately computed and then added. On the other hand, CTP2's gradient involves `cross' terms, such as $-\frac{y}{\phi_2(\theta)} \cdot \nabla_\theta \phi_N(\theta)$, which could potentially provide a more coherent gradient update.
>
> Further, assume both $\phi_{N}$ and $\phi_{K}$ are noisy estimates of an underlying true signal $$\phi_N(\theta) = \phi + \epsilon_N, \quad \phi_K(\theta) = \phi + \epsilon_K,$$ with independent zero-mean terms $ \epsilon_N, \epsilon_K $ each with variance $\sigma^2$. Assume
> $$\nabla_\theta \phi_N(\theta) \approx \nabla_\theta \phi_K(\theta) \equiv \nabla_\theta \phi.$$
> Define $$g(\phi) = -\frac{y}{\phi} + \frac{1-y}{1-\phi} \quad \text{and} \quad g'(\phi) = \frac{y}{\phi^2} + \frac{1-y}{(1-\phi)^2}.$$
> Using a first-order Taylor expansion, we can approximate$$g\bigl(\phi_N(\theta)\bigr) \approx g(\phi) + g'(\phi) \epsilon_N, \quad g\bigl(\phi_K(\theta)\bigr) \approx g(\phi) + g'(\phi) \epsilon_K. $$ The gradient for CTP1 becomes:
> $$\nabla_\theta \mathcal{L_1}\approx 2g(\phi)\nabla_\theta \phi+g'(\phi)(\epsilon_N+\epsilon_K)\nabla_\theta\phi.$$
> The noise term is $\Delta_1 = g'(\phi)(\epsilon_N+\epsilon_K) \nabla_\theta \phi,$ with variance
> $$\operatorname{Var}[\Delta_1] \propto \left(g'(\phi)\right)^2\operatorname{Var}(\epsilon_N+\epsilon_K) = \left(g'(\phi)\right)^2 \cdot 2\sigma^2.\quad(3)$$
> Similarly, for CTP2, the noise term is
> $$\Delta_2 = g'(\phi)\frac{\epsilon_N+\epsilon_K}{2} \nabla_\theta \phi, $$
> with variance
> $$\operatorname{Var}[\Delta_2] \propto \left(g'(\phi)\right)^2\operatorname{Var}\left(\frac{\epsilon_N+\epsilon_K}{2}\right) = \left(g'(\phi)\right)^2\frac{1}{4}\cdot 2\sigma^2 = \left(g'(\phi)\right)^2\frac{\sigma^2}{2}.\quad(4)$$
> By comparing Eq.3 and Eq.4 we can see the noise variance for CTP2 is reduced by a factor of 4 compared to CTP1's. This reduction could lead to smoother and more stable optimization dynamic, which is also what we observed empirically.

---

> > ### Comment · Reviewer_ck5p · 2025-04-03
> >
> > Dear authors,
> >
> > Thank you for the rebuttal. I find most of my concerns have already been addressed. I will keep the positive score.
> >
> > Best,

---

> > > ### Author Response · Authors · 2025-04-04
> > >
> > > Dear Reviewer ck5p,
> > >
> > > Thank you very much for your feedback and for keeping the score. We truly appreciate your support and encouragement. Your positive evaluation of our work means a great deal to us, and we are grateful for your time and thoughtful review.
> > >
> > > We look forward to any further suggestions you may have in the future.
> > >
> > > Best regards,
> > >
> > > Authors of the Paper 7442

---

### Official Review · Reviewer_H1qZ · 2025-03-13

**Overall Recommendation:** 2

**Summary:**

This paper proposes integrating Knowledge Graph Embedding methods into Neural Theorem Provers to address challenges in optimization and efficiency.

**Claims And Evidence:**

The claims made in the paper are well-supported by evidence.

**Essential References Not Discussed:**

No

**Experimental Designs Or Analyses:**

The authors provided detailed descriptions of their experimental designs, including dataset selection, model implementation, and hyperparameter tuning. The results comprehensively demonstrate the performance differences among various methods.

**Methods And Evaluation Criteria:**

The proposed methods and evaluation criteria are appropriate for the problem at hand.

**Other Comments Or Suggestions:**

1. Conduct a thorough analysis of why different KGE methods perform variably in CTP3 and CTP4, and propose corresponding improvements.

2. Include a comparative analysis with multi-modal knowledge graph reasoning methods to further highlight the innovations and advantages of the proposed approach.

**Other Strengths And Weaknesses:**

Strengths:
1. The paper proposes an innovative framework for integrating KGE methods with NTPs to address the optimization challenges of NTPs.

2. The experimental results demonstrate the effectiveness and applicability of the proposed methods across multiple datasets.


Weaknesses:
1. The reasons for the accuracy drop in CTP3 and CTP4 on large-scale datasets were not thoroughly analyzed.

2. The relationship with emerging multi-modal knowledge graph reasoning methods was not fully discussed, potentially limiting the understanding of the paper's innovation and limitations.

**Questions For Authors:**

1. What are the main reasons for the significant performance differences of different KGE methods in CTP3 and CTP4?

2. How do you view the relationship between this paper's approach and multi-modal knowledge graph reasoning methods? Is there potential for further integration?

**Relation To Broader Scientific Literature:**

The authors mention most related work.

**Theoretical Claims:**

The paper does not present theoretical claims that require verification. Its primary contribution lies in proposing a practical framework for integrating KGE methods with NTPs, rather than theoretical innovation. Therefore, this section is not applicable.

---

> ### Author Rebuttal · Authors · 2025-03-30
>
> > ***The reasons for the accuracy drop in CTP3 and CTP4 on large-scale datasets, and propose corresponding improvements***
>
> For CTP3, we recognize two issues that limits its performance particularly on large-scale dastasets, and propose two approaches to improve its performance. Recall for CTP3 we replace top-$k$ retrieval by using translational KGEs ($e.g.$ TransE) to directly compute the unknown entity, which means we are effectively doing top-1 retrieval. This results in two main drawbacks: 1. Limited expressiveness due to the top-1 retrieval. 2. Susceptible to spurious proof paths where subject and object are connected but logically irrelevant. For example, consider the path $(s, born\_in, o_1)$, $(o_1, located\_in, o_2)$, $(o_2, notable\_people, o)$, where $s$ and $o$ are connected but irrelevant. These issues become more obvious when the size and complexity of the dataset increase.
>
> To mitigate the above issues, we additionally propose two methods.
>
> **Filtering spurious relation paths.** We consider using the Path Constraint Resource Allocation (PCRA) for calculating the path reliability by measuring how much resource flows from the head to the tail entity following a path, as inspired by PTransE [1].
>
> Formally, for a pair of $s$ and $o$ and a path $p = (r_1, r_2, \dots, r_n)$, the flow path can be written as $s \xrightarrow{r_1} S_1 \xrightarrow{r_2} \dots \xrightarrow{r_n} S_n$, where $S_i$ are sets of entities, and $o \in S_n$. Following the notation in PTransE, given any entity $m \in S_i$, the resource flowing to $m$ is defined as
>
> $$R_p(m) = \sum_{n \in S_{i-1}(\cdot, m)} \frac{1}{|S_i(n, \cdot)|}R_p(n),$$
>
> where $S_{i-1}(\cdot,m)$ are its direct predecessors in $S_{i-1}$, and $S_i(n, \cdot)$ is the direct successors of $n \in S_{i-1}$. By calculating $R_p(m)$ recursively from $s$ to $o$, we can obtain the final resource (reliability) of the path $p$ given $s$ and $o$. For more details please refer to~\cite{ptranse}. During training, we then mask out the path with lowest 10\% reliability score (we do not modify for evaluation stage).
>
> **Learnable entity expansion module.** To alleviate the issue with top-1 retrieval, we consider adding an additional learnable module which expands the resulting entity embedding obtained from the translational KGE function to $k$ neighboring entities. In other words, we learn a  linear layer $\mathbf{W}$ with shape $(d, k)$ to encode a set of related entities $\\{S\\}_k$ given the calculated entity $s$, $i.e.$ $p(\\{S\\}_k|s, \mathbf{W})$.
>
> In the table below we show the results for incorporating PCRA and Entity Expansion (EE) module with $k=10$ for CTP3. We can observe noticeable improvements, especially on larger datasets such as FB122 and WN18RR.
>
> ||UMLS|Kinship|Nations|FB122|WN18RR|Codex-s|
> |-|-|-|-|-|-|-|
> |MRR |0.65|0.5|0.53|0.53|0.37|0.35|
> |w/ PCRA|0.66|0.50|0.54|0.55|0.39|0.38|
> |w/ EE|0.67|0.52|0.54|0.60|0.41|0.42|
> |w/ both|0.67|0.52|0.55|0.63|0.42|0.44|
>
> For CTP4 the performance drop on larger dataset is mainly because of lack of negative samples during CTP training. Since in CTP4 we utilize KGE to rank entities, we rely heavily on the quality of the conditional probability $p(s,r|o)$ learned through the KGE function. This is discussed in Section A.5 and Table 11 of the appendix. We show that by increasing the number of negative samples, we can dramatically improve the performance of CTP4 to be close to CTP (<0.01 difference on HITS@10 on FB122) despite being 1000x faster during evaluation.
>
> [1] Modeling Relation Paths for Representation Learning of Knowledge Bases, ACL 2015
> > ***Conduct a thorough analysis of why different KGE methods perform variably in CTP3 and CTP4***
>
> We provide an explanation on why similarity-based KGE methods work better with NTPs than translational KGEs because of the better alignment in their score function. Due to space limit, please refer to the section named *2.why using similarity-based KGEs can achieve better performance* in our reply to Reviewer Bi1c.
>
> Additionally, we provide a theoretical justification for the better performance of CTP2 over CTP1 under the *Theoretical justification CTP1 vs. CTP2.* section under our reply for Reviewer ck5p.
>
> > ***Include a comparative analysis with multi-modal knowledge graph reasoning methods. How do you view the relationship between this paper's approach and multi-modal knowledge graph reasoning methods?***
>
> We would like to clarify that we mentioned multi-modal KG reasoning in the introduction because it is one of our main motivations to study NTP and is our on-going research, yet it is **not** the focus in this work. Therefore, we feel it might be less coherent to conduct analysis regarding multi-modal KG reasoning in this work.
>
> However, since our proposed integration (in particular, CTP2) is a plug-and-play approach without introducing new parameters or significantly modifying model architecture, we believe it should be agnostic to the original modality behind the embedding.

---

### Official Review · Reviewer_8H6v · 2025-03-14

**Overall Recommendation:** 3

**Summary:**

The paper proposes to integrate Knowledge Graph Embedding (KGE) methods with Neural Theorem Provers (NTPs) to enhance neuro-symbolic reasonings, and hence. The author proposes 4 ways to use KGEs and explain the methodology, the most intuitive variant seems to be use the kge at each proof step. Then the paper uses extensive experiments to show the performance of newly proposed methodology in different datasets, the ablation studies is also provided.

**Claims And Evidence:**

Generally speaking, the paper provides empirical results as evidence for their claims.

**Essential References Not Discussed:**

No.

**Experimental Designs Or Analyses:**

I think it's valid.

**Methods And Evaluation Criteria:**

Yes.

**Other Comments Or Suggestions:**

I recommend adding more grounded examples for illustrations at the start of the paper instead of just Figure 1.

**Other Strengths And Weaknesses:**

Strengths

1.	The approach to integrate KGEs to improve the NTPs is novel and reasonable for advancing the capabilities of neuro-symbolic systems. The four variants of CTPs shows the paper also offers detailed analysis on how to utilize the KGEs.

2.	The paper provides extensive experimental results that not only compare them against a variety of benchmarks but also includes ablation study and case-study for detailed discussion of the empirical results.

Weaknesses

1.	The paper is dense with technical details as one can observe the paper uses a lot of spaces to talk about existing research and methods, make it less accessible to readers not familiar with the specific fields of NTPs or KGEs. In contrast, the newly proposed CTP may require more room for further illustrations.

2.	The improvement of the method seems to be marginal, as the CTP generally falls behind previous methods like NBF-net, though the paper has argued it has stronger efficiency.

**Questions For Authors:**

Why NBFNet is not included in Table 5.


Thanks author, I have read your response and changed my ratings corrspondingly.

**Relation To Broader Scientific Literature:**

The paper studies Neural Theorem Provers.

**Theoretical Claims:**

This paper has no proofs.

---

> ### Author Rebuttal · Authors · 2025-03-30
>
> > ***The paper is dense with technical details as one can observe the paper uses a lot of spaces to talk about existing research and methods, make it less accessible to readers not familiar with the specific fields of NTPs or KGEs. In contrast, the newly proposed CTP may require more room for further illustrations.***
>
> Thank you for your advice. We will incorporate more description on CTP and modify the introduction/method section to be easier to follow.
>
> > ***The improvement of the method seems to be marginal, as the CTP generally falls behind previous methods like NBF-net, though the paper has argued it has stronger efficiency.***
>
> First, out of the seven Knowledge Graph Completion datasets tested, we are only slightly behind NBFNet for 2 of them (Kinship and WN18RR). While CTP2 lags behind NBFNet on Kinship for 0.09 MRR, we note that Kinship is a special case where CTP2 has little effect as the dataset has little multi-hop proof paths, yet for CTP2 we specifically add KGE priors along the proof path. In addition, we show that CTP2 clearly outperforms NBFNet and all the other baselines on Cluttr dataset (Table 4).
>
> Second, we show that by incorporating KGE priors, we can achieve consistent performance improvement across most if not all (excluding Kinship) datasets (CTP2), without introducing any new parameters and with negligible additional computational overhead. While the improvements of CTP2 over baseline CTP on small statistical learning datasets (e.g. Nations, UMLS) are less significant (since the baseline CTP’s accuracy on these datasets is already relatively high). The improvement becomes much more obvious on WN18RR and CodEx, with an improvement on MRR of 0.07 and 0.16, respectively.
>
> Lastly, NBFNet has been a SOTA KGC method, and many more recent KBC works’ performances also do not surpass it (e.g. DiffLogic, LERP).
>
> > ***I recommend adding more grounded examples for illustrations at the start of the paper instead of just Figure 1.***
>
> Thank you for the suggestion! We will modify the manuscript to incorporate a grounded CTP inference illustration.
>
> > ***Why NBFNet is not included in Table 5.***
>
> Thank you for the question. We initially tried to run NBFNet on FB122 with various hyperparameters but the accuracy was all low compared to its performance on other datasets and we were uncertain if we missed something, therefore we did not include it.
>
> Below we show results on FB122 with NBFNet included. Additionally, we rerun Table 5 with larger retrieval size ($k=128$). This was done on WN18RR and CodEx datasets (L648-653) but not FB122 because the baseline GNTP was run on FB122 with $k=10$, and we want to draw a fair comparison with it. Due to space limit, we are only showing the Test-ALL split of FB122.
>
> ||| H@3  | H@5  | H@10 | MRR  |
> |-|----------------------|------|------|------|------|
> |With Rules||||||
> || KALE-P               | 61.2 | 66.4 | 72.8 | 0.52 |
> || KALE-J               | 70.7 | 73.1 | 75.2 | 0.67 |
> || ASR-D                | 71.7 | 73.6 | 75.7 | 0.67 |
> || KBLRN                | 74.0 | 77.0 | 79.7 | 0.70 |
> |Without Rules||||||
> || TransE               | 58.9 | 64.2 | 70.2 | 0.48 |
> || DistMult             | 67.4 | 70.2 | 72.9 | 0.63 |
> || ComplEx            | 67.3 | 69.5 | 0.72 | 0.64 |
> || GNTP                 | 61.5 | 63.2 | 64.5 | 0.61 |
> || CTP                  | 69.1 | 70.5 | 71.2 | 0.68 |
> || NBFNet             | 57.2 | 59.6 | 70.6 | 0.51 |
> |||||||
> || CTP₁                 | 65.4 | 64.3 | 65.0 | 0.62 |
> || CTP₂                 | **76.1** | **76.4**| **78.3** | **0.75** |
> || CTP₃                 | 59.4 | 60.8 | 62.2 | 0.53 |
> || CTP₄                 | 65.7 | 66.3 | 69.6 | 0.64 |
>
> As we can see, CTP2 achieves the best accuracy overall. NBFNet's accuracy, however, is significantly lower. Due to time limit we are not able to further search for hyperparameters as it takes more than a day to train NBFNet on FB122 with a V100 GPU. However we will keep experimenting and will update in the followup reply if we find better accuracy with NBFNet.
>
> > ***Replies to other reviewers***
>
> Here we would like to refer to some questions from other reviewers and our replies that could potentially be of your interest.
>
> **Theoretical justification of why CTP2 is better.** We provide a theoretical justification for the better performance of CTP2 over CTP1 under the *Theoretical justification CTP1 vs. CTP2.* section under our reply for Reviewer ck5p.
>
> **Explanation on why similarity-based KGEs perform better than translational KGEs on NTPs.** Please refer to the section named *why using similarity-based KGEs can achieve better performance* in our reply to Reviewer Bi1c.
>
> **Explanation on why CTP3-4's accuracy drop sharply on large-scale dataset, and methods of improvement.** Please see our reply to Reviewer H1qZ, section *The reasons for the accuracy drop in CTP3 and CTP4 on large-scale datasets, and propose corresponding improvements*.

---

### Official Review · Reviewer_Bi1c · 2025-03-15

**Overall Recommendation:** 3

**Summary:**

This paper investigates the integration of Neural Theorem Provers (NTPs) and Knowledge Graph Embeddings (KGEs) to enhance soft unification and reasoning efficiency. The paper systematically explores four strategies for integrating KGEs into NTPs:
CTP1: Uses KGE as an auxiliary loss to support NTP training.
CTP2: Injects KGE-based similarity scores score function into NTPs as an auxiliary score.
CTP3: Replaces the topk retrieval with a translation-based operation to improve inference efficiency.
CTP4: Replaces the final step of NTP’s proof evaluation with KGE-based ranking to  reduce computational overhead.

The paper demonstrates that these integrations can substantially improve NTPs in both accuracy and computational efficiency. Additionally, the paper provides detailed ablation studies. This work is the first systematic study of KGE integration into NTPs.

**Claims And Evidence:**

The paper makes several key claims, including:
1. Leveraging the properties of KGEs can drastically improve the inference and evaluation efficiency of NTPs. The paper reports that CTP2 achieves higher accuracy than existing NTP-based methods on multiple datasets, including FB122, Nations, and UMLS.

2. CTP3 and CTP4 significantly improve computational efficiency. This claim is well supported by experiments. The paper provides empirical runtime comparisons, demonstrating substantial speed improvements. However, a formal complexity analysis is missing.

3. This is the first systematic study of integrating KGE into NTP. The paper “End-to-End Differentiable Proving” has proposed the method that integrates KGEs into NTPs, but this paper is the first systematic study for the integration of KGEs into NTPs.

**Essential References Not Discussed:**

To the best of my knowledge, the paper discusses the most relevant prior work, and I did not identify essential references that are missing.

**Experimental Designs Or Analyses:**

The experimental using multiple benchmark datasets and standard metrics, including comparisons with baselines, efficiency evaluation, visualization, hyperparameter analysis and effect of using different KGEs. The experimental design is generally sound, lacks theoretical analysis for performance differences. In addition, there appears to be an error in the bolding of values in Table 2. For the HITS@10 metric on UMLS, the best result should be CTP1 rather than CTP2.

**Methods And Evaluation Criteria:**

Methods: The proposed methods which integrate KGEs with NTPs through different distinct strategies (CTP1–CTP4)—are well-motivated by the limitations of NTPs, such as the poorly  structured embedding space. The integration choices are reasonable, as they directly address these issues by leveraging KGE’s structured representation learning. However, a more formal theoretical justification for why CTP2 work better than others is missing.

Evaluation Criteria:
The evaluation is conducted on standard knowledge graph reasoning benchmarks, including FB122, WN18RR, Nations, UMLS, Kinship, CLUTRR. These datasets are appropriate choices for evaluating reasoning tasks. The paper evaluates performance using standard metrics: Mean Reciprocal Rank (MRR) and Hits@m, which are widely used for knowledge graph completion tasks.

**Other Comments Or Suggestions:**

1. There appears to be an error in the bolding of values in Table 2. For the HITS@10 metric on UMLS, the best result should be CTP1 rather than CTP2.
2. The capitalization of model names such as ComplEx and RotatE is inconsistent in the paper. For example, "COMPLEX" and "ComplEx" are both used.

**Other Strengths And Weaknesses:**

Strengths：
1. The paper is well-written and clearly structured, making the key contributions and technical details easy to follow. The motivation for integrating KGE with NTPs is well-explained.
2. The authors have conducted comprehensive experiments, including comparisons with baselines, efficiency evaluation, visualization, hyperparameter analysis and effect of using different KGEs.

Weaknesses:
1. There is no theoretical analysis for performance differences. Specifically, the experimental results indicate that CTP2 achieves the best performance, but the paper does not explicitly discuss why CTP2 can achieve the best performance? Can it be theoretically explained?
2. The “Effect of using different KGEs” part of experimental results indicate that similarity-based KGEs, ComplEx DistMult generally yields the best performance, whereas translation-based KGE TransE and RotatE often lag back by a large margin. The paper lacks an explanation for this phenomenon. Would it be possible to further analyze why using similarity-based KGEs can achieve better performance?

**Questions For Authors:**

Please see the weaknesses above.

**Relation To Broader Scientific Literature:**

This paper is the first systematic study for the integration of KGEs into NTPs and propose four integration strategies, improving both performance and efficiency. Using KGE as auxiliary was originally proposed in “End-to-End Differentiable Proving”, but it was only briefly mentioned without any further exploration, and is not used in any subsequent NTPs. Compared with NTPs: NTP, GNTP, CTP, and link predictors: ComplEx, DistMult, NEURALLP, MINERVA, DIFFLOGIC, LERP and GNN NBFNET, the method proposed in this paper achieved better results in most cases.

**Theoretical Claims:**

The paper does not provide formal theoretical proofs regarding the correctness of the proposed methods. The paper relies on experimental evidence to support its claims.

---

> ### Author Rebuttal · Authors · 2025-03-30
>
> > ***Complexity Analysis:***
>
> Below we provide complexity analysis for the baseline CTP and the proposed variants. We particularly focus on the final step at each proof path during evaluation, since it is the computational bottleneck in the NTP framework.
>
> Let $|\mathcal{E}|$ be the number of entities, $k$ be the number of top facts retrieved per $(s, r,t)$ pair, $d$ be the embedding dimension, $N$ be the number of proof path templates, $D$ be the depth of the recursive proof tree.
>
> **Baseline CTP** retrieves top-k facts for each combination of the missing entity and the known predicate-object/subject pair, followed by the unification score calculation between two tensors of shape $(|\mathcal{E}|, k, 3d)$. **Retrieval:** we use the IndexFlatL2 index from FAISS library, which has linear complexity $w.r.t.$ number of items. Since we are retrieving for each entity combination, the complexity of retrieval is $O(|\mathcal{E}|^2)$ for the final step at each proof path. **Score calculation:** we employ the Gaussian RBF kernel as the similarity metric between two embeddings. The RBF kernel is defined as
>
> \begin{equation}\text{RBF}(x_i, y_i) = \exp\big(-\frac{||x_i-y_i||^2}{2\sigma^2}\big),\end{equation} which has approximately linear complexity $w.r.t.$ number of entities $|\mathcal{E}|$ and dimension $d$, $i.e.$ $O(|\mathcal{E}| \cdot kd)$.
>
> Since this has to be done at each individual proof path, the complexity becomes $O(ND(\mathcal{|E|}^2 + \mathcal{|E|} \cdot kd))$.
>
> **$\textbf{CTP}_1$** adds the KGE objective only as a loss term. Therefore the evaluation complexity is the same as baseline $\text{CTP}_1$, which is $O(ND(\mathcal{|E|}^2 + \mathcal{|E|} \cdot kd))$.
>
> **$\textbf{CTP}_2$** adds the KGE scoring function on top of baseline CTP's existing similarity function (RBF kernel). However, as we find the additional KGE scoring function is only necessary during training and can be omitted during evaluation time, the evaluation complexity is the same as baseline $\text{CTP}_1$.
>
> **$\textbf{CTP}_3$** replaces retrievals with translational KGEs to directly compute the next unknown tail entity. This reduces the retrieval complexity from quadratic to linear $w.r.t. \mathcal{|E|}$. Therefore the complexity becomes $O(ND(\mathcal{|E|} + \mathcal{|E|} \cdot kd))$.
>
> **$\textbf{CTP}_4$** replaces the final ranking step from baseline CTP's retrieval & unification score calculation with direct ranking based on the KGE score. Assume a linear complexity for the KGE function $w.r.t.$ number of entities ($e.g.$ ComplEx), the overall complexity is further reduced to $O(ND(\mathcal{|E|})$.
>
> > ***1.Theoretical analysis for why CTP2 is better***
>
> We provide a theoretical analysis from the gradient perspective comparing CTP2 to CTP1 (since they are similar to each other compared to CTP3-4). Due to space limit, we kindly ask the reviewer to refer to the section named **Theoretical justification CTP1 vs. CTP2.** in our reply to Reviewer ck5p. We are only able to show essential steps in the derivation here, but we will add the full derivation to our manuscript.
>
> > ***2.why using similarity-based KGEs can achieve better performance***
>
> We conjecture the score function of similarity-based KGEs align better with the similarity metric (RBF Kernel) used in NTP, as compared to translational KGEs, which use the negative distance between original and translated entity embedding as the score.
>
> Specifically, similarity-based KGEs like ComplEx or DistMult generally use an inner product (or bilinear form) $\langle \cdot, \cdot \rangle$ to score triplets:
>
> $$s_{\text{sim}} (s,r,o) = \langle (s,r), o \rangle,$$
>
> while translational-based KGEs (such as TransE) use the distance as metric
>
> $$s_{\text{transE}} = -||s+r-o||.$$
>
> On the other hand, since RBF kernel is defined as
>
> $$\text{RBF}(x, y) = \exp\big(-\frac{||x-y||^2}{2\sigma^2}\big).$$
>
> For simplification, assume the embeddings are normalized, we have
>
> $$||x-y||^2 \approx 2 - 2\langle x, y \rangle,$$
>
> therefore
>
> $$\text{RBF}(x_i, y_i) = \exp\big(-\frac{1 - \langle x , y \rangle}{\sigma^2}\big).$$
>
> This means RBF kernel is essentially a monotonic function of the inner product $\langle x, y \rangle$, and therefore the gradient of the similarity-based KGE score is directly tied to inner product-based KGEs ($e.g.$ ComplEx and DistMult). In experiments, we also observe faster convergence and higher accuracy when using ComplEx and DistMult.
>
> > ***Other Comments: bolding and capitalization errors***
>
> Thank you for pointing them out! We will correct them in our final manuscript.

---

> > ### Comment · Reviewer_Bi1c · 2025-04-06
> >
> > Thanks for solving my concerns, and I will keep my original score.

---

> > > ### Author Response · Authors · 2025-04-07
> > >
> > > Dear Reviewer Bi1c,
> > >
> > > Thank you very much for your feedback and for keeping the score. We truly appreciate your support and encouragement. Your positive evaluation of our work means a great deal to us, and we are grateful for your time and thoughtful review.
> > >
> > > We look forward to any further suggestions you may have in the future.
> > >
> > > Best regards,
> > >
> > > Authors of the Paper 7442

---

### Decision · Program_Chairs · 2025-05-01

**Decision:**

Accept (poster)

**Comment:**

This paper focuses on the integration of NTPs and KGEs for improving soft unification. The paper proposes several different strategies (the high-level idea of these strategies appear in the literature but the devil is often in the details of integration). The paper also provides an extensive empirical evaluation.

one of the concerns of some of the reviewers was lack of theoretical explanation but it is understandable that theoretical justification don't come easy and it would be a loss for the community to not know the strategies that work (at least empirically).

Given the extensive empirical evaluation, I recommend acceptance.